# BREAKING SAFETY PARADOX WITH FEASIBLE DUAL POLICY ITERATION

**Yujie Yang**[*1]**, Jinglin Teh**[*1]**, Ziyu Lin**[1]**, Kaicheng Yu**[2]**, Tao Zhang**[1,3]**, Shengbo Eben Li**[†1]
[1]Tsinghua University, [2]Westlake University, [3]SunRisingAI Lab
{yangyj21,zhengjl24}@mails.tsinghua.edu.cn, lin_ziyu@cau.edu.cn,
kyu@westlake.edu.cn, zhang.t1983@gmail.com, lishbo@tsinghua.edu.cn

## ABSTRACT

Achieving zero constraint violations in safe reinforcement learning poses a significant challenge. We discover a key obstacle called the safety paradox, where improving policy safety reduces the frequency of constraint-violating samples, thereby impairing feasibility function estimation and ultimately undermining policy safety. We theoretically prove that the estimation error bound of the feasibility function increases as the proportion of violating samples decreases. To overcome the safety paradox, we propose an algorithm called feasible dual policy iteration (FDPI), which employs an additional policy to strategically maximize constraint violations while staying close to the original policy. Samples from both policies are combined for training, with data distribution corrected by importance sampling. Extensive experiments show FDPI's state-of-the-art performance on the Safety-Gymnasium benchmark, achieving the lowest violation and competitive-to-best return simultaneously.

## 1 INTRODUCTION

Reinforcement learning (RL) has achieved promising performance in many challenging tasks such as video games (Vinyals et al., 2019), board games (Schrittwieser et al., 2020), autonomous driving (Wurman et al., 2022), and drone racing (Kaufmann et al., 2023). RL solves an optimal control problem by finding a policy that maximizes the expected cumulative rewards. However, real-world control tasks often demand more than reward maximization—they require strict adherence to safety constraints, where even rare violations can lead to catastrophic outcomes. Achieving zero constraint violations in these tasks remains a significant challenge.

A key element in safe RL is the feasibility function, which evaluates whether a state can satisfy safety constraints over an infinite horizon. This function not only defines the feasible region of a policy but also serves as a safety-oriented learning target. Examples of feasibility functions include cost value function (CVF) (Altman, 2021), Hamilton-Jacobi (HJ) reachability function (Bansal et al., 2017), and constraint decay function (CDF) (Yang et al., 2023b). These functions are typically learned through fixed-point iteration based on their risky self-consistency conditions (Li, 2023; Yang et al., 2024)—analogous to the Bellman equation for value functions. These conditions establish recursive relationships between temporally adjacent states, allowing feasibility functions to capture long-term safety at all states.

While learning-based feasibility functions are crucial in ensuring safety, we discover that they inherently prevent policies from achieving zero violations due to a phenomenon we term the safety paradox. Our analysis reveals that as policy safety improves and violating samples become sparser, the estimation error of the feasibility function increases. This impairs the accuracy of the feasible region and introduces bias into the policy's learning target, ultimately undermining safety performance. This phenomenon differs fundamentally from the sparse reward problem in standard RL, where achieving higher rewards directly facilitates further reward improvement. In contrast, the safety paradox forms a self-defeating cycle where improving safety degrades the learning conditions for further safety optimization.

---

*Equal contribution

†Corresponding author

Existing methods for addressing sample sparsity, which we categorize as passive and active, are inadequate for resolving the safety paradox. Passive methods such as prioritized experience replay (PER) (Schaul et al., 2015) reweight samples in the replay buffer to emphasize critical transitions. However, their efficacy is limited when critical samples are inherently rare, and they fail to break the safety paradox's self-defeating cycle. Active methods such as curiosity-driven exploration (Pathak et al., 2017) modify the environment or agent behavior to generate critical samples. While potentially more effective, these methods induce behavioral shifts that can steer the policy away from optimality, and their implementation often requires intrusive task modifications, which may not be feasible in practice.

In this paper, we propose an algorithm called feasible dual policy iteration (FDPI), which breaks the safety paradox by incorporating an additional dual policy designed to maximize constraint violations. This approach effectively increases the proportion of constraint-violating samples without increasing the total number of samples, thereby reducing feasibility function estimation error and pushing policy safety to a higher level. A challenge of this approach is the distributional shift that occurs when combining data from both policies. We address this through an importance sampling (IS) scheme that approximates the marginal state distribution with a truncated trajectory distribution. We further introduce KL divergence constraints between the two policies to ensure numerical stability of IS. Extensive experiments on the Safety-Gymnasium benchmark demonstrate FDPI's state-of-the-art performance.

## 2 RELATED WORK

**Safe RL algorithms**  A prominent class of safe RL algorithms is called iterative unconstrained RL, which transforms the safe RL problem into a series of unconstrained RL problems, typically via the method of Lagrange multipliers (Paternain et al., 2019). Under this framework, researchers explored different kinds of feasibility functions, including CVF (Chow et al., 2018; Tessler et al., 2018), HJ reachability function (Yu et al., 2022; 2023), and control barrier function (Yang et al., 2023a;b). Another class is called constrained policy optimization, which incorporates safety constraints in each iteration of policy optimization. The most representative example is CPO (Achiam et al., 2017), which adopts a trust region update with linearized objective and constraints. Several improvements over CPO have been proposed, including projection methods (Yang et al., 2020; 2022) and first-order methods (Zhang et al., 2020; 2022). For finite-horizon problems, Zhao et al. (2023) and Zhao et al. (2024) convert state-wise constraints to cumulative constraints and bound the worst-case violation. A common practice of these algorithms is to estimate feasibility functions from sampled data.

**Critical sample augmentation in RL**  There are two kinds of methods to increase critical samples in RL: passive methods and active methods. Passive methods focus on biasing the replaying process to prioritize experiences that are likely to be more informative for learning. A representative example is PER (Schaul et al., 2015), which replays samples with larger temporal difference (TD) errors more frequently. Other methods include prioritizing similar experiences to the current policy (Novati & Koumoutsakos, 2019) and modifying certain information in replayed samples (Andrychowicz et al., 2017). Active methods involve modifying the environment or the agent's behavior to deliberately generate critical samples. Some algorithms employ adversary policies to generate challenging scenarios (Pinto et al., 2017; Feng et al., 2023), while others use auxiliary rewards to guide exploration (Jaderberg et al., 2016; Pathak et al., 2017). Unlike these methods, our algorithm requires no environment modifications or reward shaping, maintaining the integrity of the original task. There are also methods that address the sample sparsity challenge through representation learning, which are discussed in Appendix D.1.

## 3 PROBLEM STATEMENT

Safe RL addresses control problems in which an agent aims to maximize long-term rewards while strictly adhering to safety constraints at every step. We consider a Markov decision process (MDP) $(\mathcal{X}, \mathcal{U}, p_{\text{init}}, P, r, \gamma)$, where $\mathcal{X} \subseteq \mathbb{R}^n$ is the state space, $\mathcal{U} \subseteq \mathbb{R}^m$ is the action space, $p_{\text{init}} \in \Delta\mathcal{X}$ is the initial state distribution, $P : \mathcal{X} \times \mathcal{U} \to \Delta\mathcal{X}$ is the transition probability, $r : \mathcal{X} \times \mathcal{U} \to \mathbb{R}$ is the reward function, and $0 < \gamma < 1$ is the discount factor. We consider a stochastic policy $\pi : \mathcal{X} \to \Delta\mathcal{U}$, whose state-value function is defined as $V^\pi(x) = \mathbb{E}\left[\sum_{t=0}^{\infty} \gamma^t r(x_t, u_t) | x_0 = x\right]$. Safety is specified

through a state constraint expressed as an inequality $h(x) < 0$, where $h : \mathcal{X} \rightarrow \mathbb{R}$ is the constraint function. The state constraint must be satisfied at every step:

$$h(x_t) \leq 0, \quad \forall\, t \geq 0. \tag{1}$$

The goal of safe RL is to find a policy that maximizes the state-value function while satisfying the state constraints over an infinite horizon.

### 3.1 FEASIBILITY IN SAFE REINFORCEMENT LEARNING

Feasibility is a core concept in safe RL that describes the long-term safety of a state. To formally define feasibility, we first introduce the reachable set.

**Definition 1** (Reachable set). *The reachable set of a policy $\pi$ from a state $x \in \mathcal{X}$, denoted $\mathcal{R}^\pi(x)$, is the set of states that can be reached with non-zero probability under $\pi$ in finite time:*

$$\mathcal{R}^\pi(x) = \{x' \in \mathcal{X} | \exists t \geq 0,\ s.t.\ P(x_t = x' | x, \pi) > 0\}, \tag{2}$$

*where $P(x_t = x' | x, \pi)$ is the probability of reaching $x'$ at time $t$ starting from $x$ and following $\pi$.*

The reachable set includes all states that will possibly be visited by $\pi$ given an initial state. Feasibility is defined based on whether all states in the reachable set is constraint-satisfying.

**Definition 2** (Feasible region). *The feasible region of a policy $\pi$, denoted $\mathrm{X}^\pi$, is the set of states from which every reachable set under $\pi$ satisfies the safety constraint:*

$$\mathrm{X}^\pi = \{x \in \mathcal{X} | \forall x' \in \mathcal{R}^\pi(x), h(x') \leq 0\}. \tag{3}$$

In safe RL, we need to find a policy whose feasible region includes all possible initial states. This requirement can be expressed as a single constraint by the feasibility function.

**Definition 3** (Feasibility function). *Function $F^\pi : \mathcal{X} \rightarrow \mathbb{R}$ is a feasibility function of $\pi$ if its zero-sublevel set equals the feasible region of $\pi$, i.e., $\{x \in \mathcal{X} | F^\pi(x) \leq 0\} = \mathrm{X}^\pi$.*

An example of a feasibility function is the CDF (Yang et al., 2023b).

**Definition 4** (Constraint decay function). *The CDF of a policy $\pi$ is defined as*

$$F^\pi(x) = \mathbb{E}_{\tau \sim \pi} \left[ \gamma^{N(\tau)} \big| x_0 = x \right], \tag{4}$$

*where $\gamma \in (0, 1)$ is the discount factor, $\tau = \{x_0, u_1, x_1, u_1, \dots\}$ is a trajectory sampled by $\pi$, and $N(\tau) \in \mathbb{N} \cup \{+\infty\}$ is the time step of the first constraint violation in $\tau$.*

In the rest of this paper, we use CDF as a concrete example of a feasibility function. However, our analysis also applies to other feasibility functions with similar properties such as CVF. The feasibility function is also called a constraint aggregation function (Yang et al., 2024) because we can replace the original infinitely many constraints (1) with a single one expressed by the feasibility function, leading to the following safe RL problem:

$$\max_\pi\ \mathbb{E}_{x \sim p_{\text{init}}}[V^\pi(x)] \quad \text{s.t.}\ \mathbb{E}_{x \sim p_{\text{init}}}[F^\pi(x)] \leq 0. \tag{5}$$

## 4 SAFETY PARADOX

A core problem in safe RL is to estimate the feasibility function. We discover that as the policy becomes safer, the estimation error of the feasibility function tends to increase. This makes the identified feasible region less accurate, which, in turn, harms policy update and deteriorates policy safety. This phenomenon is called the safety paradox.

### 4.1 ESTIMATION ERROR BOUND OF CDF

In safe RL, the CDF is computed by solving its risky self-consistency condition with fixed-point iteration (Yang et al., 2023b):

$$F^\pi(x) = \mathbb{E}_{x' \sim P(\cdot | x, u), u \sim \pi(\cdot | x)} \left[ c(x) + (1 - c(x)) \gamma F^\pi(x') \right], \tag{6}$$

where $c(x) = \mathbb{I}[h(x) > 0]$ is an indicator function for constraint violation. In practice, the expectation above is estimated by sample average. Equation (6) can be viewed as a one-step TD estimate of the CDF. Since TD involves bootstrapping of the estimated CDF itself, the analysis of estimation error becomes complicated. Here, we consider a Monte Carlo (MC) estimate instead for theoretical simplicity: $\hat{F}^\pi(x) = 1/K \sum_{i=1}^{K} \gamma^{N(\tau_i)}$, where $\tau_1, \tau_2, \ldots, \tau_K$ are $K$ independent trajectories starting from $x$ sampled by $\pi$. We discuss extension to TD estimate at the end of Section 4.2.

An inaccurate CDF leads to incorrect identification of the feasible region, i.e., feasible states misidentified as infeasible and vice versa. To minimize misidentification, we must bound the estimation error. We show that the bound of the relative estimation error of CDF is related to the expectation and variance of the number of steps to the first violation. Before that, we assume that these two quantities are finite.

**Assumption 1.** *For any infeasible state $x \in \mathcal{X}$ under policy $\pi$, let $\mu_N^\pi(x) = \mathbb{E}_{\tau \sim \pi}[N(\tau)|x_0 = x]$ and $\sigma_N^{2;\pi}(x) = Var_{\tau \sim \pi}[N(\tau)|x_0 = x]$. We have $\mu_N^\pi(x) < +\infty$ and $\sigma_N^{2;\pi}(x) < +\infty$.*

**Theorem 1.** *For any infeasible state $x \in \mathcal{X}$ under policy $\pi$, let $\hat{F}^\pi(x)$ be the MC estimate of the CDF. Under Assumption 1, the expected relative estimation error is bounded by:*

$$\mathbb{E}_{\tau_1, \tau_2, \ldots, \tau_K} \left[ \left| \frac{\hat{F}^\pi(x) - F^\pi(x)}{F^\pi(x)} \right| \right] \le \frac{1}{\sqrt{K}} |\ln \gamma| \sigma_N^\pi(x) + (\ln \gamma)^2 \frac{\sigma_N^{2;\pi}(x)}{\gamma^{\mu_N^\pi(x)}}. \tag{7}$$

*Proof Sketch.* Construct two auxiliary functions $H^\pi(x) = \gamma^{\mu_N^\pi(x)}$ and $\hat{H}^\pi(x) = \gamma^{\hat{\mu}_N^\pi(x)}$, where $\hat{\mu}_N^\pi(x) = 1/K \sum_{i=1}^{K} N(\tau_i)$. Use Taylor expansion to obtain the bounds of $|F^\pi(x) - H^\pi(x)|$, $|\hat{F}^\pi(x) - \hat{H}^\pi(x)|$, and $|H^\pi(x) - \hat{H}^\pi(x)|$. The result follows by the triangle inequality. See Appendix A.1 for the complete proof. □

The number of samples $K$ in the error bound (7) is related to the batch size and is a constant throughout training. The only two variables relevant to the error bound is the expectation and variance of steps to violation. While it is obvious that the expectation increases as the policy becomes safer, how the variance changes is not easily observed and requires further analysis.

## 4.2 RELATIONSHIP BETWEEN POLICY SAFETY AND ESTIMATION ERROR BOUND

In this section, we show that under mild assumptions, the variance of steps to violation increases as the policy becomes safer. To begin with, we introduce a function to measure the "distance" to constraint violation.

**Assumption 2.** *There exists a continuous function $D : \mathcal{X} \to \mathbb{R}$, such that $\forall x \in \mathcal{X}$, $D(x) \ge 0$ and $D(x) = 0 \iff h(x) > 0$.*

Examples of $D$ include the Euclidean distance to obstacles in collision avoidance tasks, and the margin to speed limit in velocity-constrained tasks. With a distance function, we can define what a "safer" policy is.

**Definition 5.** *A policy $\pi'$ is safer than a policy $\pi$ in a state $x$ if $P(D(x'_{\pi'}) < D(x)) \le P(D(x'_\pi) < D(x))$, where $x'_\pi \sim p_\pi(\cdot|x)$ is the next state under $\pi$.*

A safer policy has a lower probability of decreasing the distance to violation. This definition naturally connects to the proportion of violating samples: if the distance to violation decreases slower, the number of steps to violation will be larger, resulting in fewer violating samples. The following assumption requires the continuity of $D$ and the system dynamics. This holds in most systems, where the changing rate of states and the control inputs are bounded.

**Assumption 3.** *The difference of $D$ between any two adjacent states is bounded by $\delta > 0$, i.e., for any $x, u, x'$ such that $p(x'|x, u) > 0$, it holds that $|D(x) - D(x')| \le \delta$.*

We split $D(x)$ into a sequence of consecutive intervals: $[D(x) - i\delta, D(x) - (i - 1)\delta)$, $i = 1, 2, \ldots, m(x) - 1$, where $m(x) = \lceil D(x)/\delta \rceil$. With Assumption 3, the agent must visit the $(i-1)$th interval before visiting the $i$th interval. We denote $N_i^\pi(x)$ as the first-visiting time from the $(i-1)$th interval to the $i$th interval. Note that $D(x)$ is not necessarily monotonic, i.e., the agent may revisit a

stage where it has visited before. $N_i^\pi(x)$ is counted until the first time the next stage is visited. The following assumption requires that for states far from violation, $N_1^\pi(x)$ is weakly correlated with the sum of the rest first-visiting times.

**Assumption 4.** *There exists $M > 0$, such that for all $x$ with $m(x) \geq M$, it holds that* $\left| Cov\left( N_1^\pi(x), \sum_{i=2}^{m(x)} N_i^\pi(x) \right) \right| \leq Var(N_1^\pi(x))/2.$

The intuition behind this assumption is that for a state far from violation, the initial step $N_1^\pi(x)$ is primarily influenced by local dynamics. In contrast, the subsequent trajectory $\sum_{i=2}^{m(x)} N_i^\pi(x)$ is governed by future stochastic events. Their correlation is weak relative to the variability of the initial step itself. With the above assumptions, we have the following theorem.

**Theorem 2.** *Let $\pi, \pi'$ be two policies. Consider a set of states $X_M \subseteq \mathcal{X}$ such that for all $x \in X_M$, (1) $x$ is infeasible under both $\pi$ and $\pi'$, (2) $m(x) \geq M$, (3) $\min\{P(D(x'_\pi) < D(x)), P(D(x'_{\pi'}) < D(x))\} \geq 0.5$. Under Assumptions 2–4, if $\pi'$ is safer than $\pi$ in all states in $X_M$, we have $\sigma_N^{2,\pi'}(x) \geq \sigma_N^{2,\pi}(x), \forall x \in X_M$.*

*Proof Sketch.* First prove that for a given policy, a state further to violation has a larger variance by stage decomposition. Then, prove that in a given state, a safer policy has a larger variance by law of total variance. Finally, extending this result to all states in $X_M$ proves the theorem. See Appendix A.2 for the complete proof. □

**Remark** The condition (3) in Theorem 2 follows from the requirement of finite mean and variance. To ensure the violation happens in finite steps, the probability of approaching the violation, i.e., $d(x') < d(x)$, must be larger than that of moving away from it in each step. Theorem 2 tells us that a safer policy (in the sense of lower probability of approaching violation) has larger variance of steps to violation in infeasible states far from violation under mild assumptions. Combined with Theorem 1, we conclude that a safer policy leads to a larger CDF estimation error bound. To help better understand this result, we give an intuitive example of a one-dimensional random walk in Appendix A.3. It is worth mentioning that although the above analysis is based on MC estimate, it can be extended to TD estimate. The core reason is that the increased variance identified in Theorem 2 propagates to the TD target. The analysis can also be extended to the CVF widely used in the constrained MDP. A detailed explanation of the extensions can be found in Appendices D.2.

## 5 FEASIBLE DUAL POLICY ITERATION

The only way to break the safety paradox is to increase constraint-violating samples. We achieve this by training an additional policy, called the dual policy, that intentionally violates the constraint.

### 5.1 COLLECTING MORE VIOLATING SAMPLES WITH A DUAL POLICY

We denote the dual policy as $\pi_d$, and for distinguishing purposes, we call the original policy the primal policy and denote it as $\pi_p$. We train a dual feasibility function $G_d(x, u) = \mathbb{E}_{\tau \sim \pi_d}[\gamma^{N(\tau)} | x_0 = x, u_0 = u]$ to help optimize the dual policy. Compared with $F_d$, $G_d$ further fixes the current action $u_0$ and computes the feasibility value starting from the state-action pair $(x, u)$. The dual policy is updated by maximizing the dual feasibility function:

$$\max_{\pi_d} \mathbb{E}_{x, u \sim \pi_d}[G_d(x, u)].$$

Both the primal policy and the dual policy are used to sample data. In practice, the sampling ratio is controlled by a hyperparameter called the dual threshold $d$, which is fixed at 0.95 in our experiments. If the running averaged proportion of feasible states is greater than $d$, the dual policy will be activated and collect half of the samples. We ensure that the total number of samples used by our algorithm, i.e., collected by both the primal and dual policies, equals that of other algorithms.

### 5.2 CORRECTING SAMPLE DISTRIBUTION VIA IMPORTANCE SAMPLING

The problem of directly using data collected by the dual policy is that the data distribution for computing the expectation in loss functions is shifted. Take the loss function of the primal feasibility

function as an example:

$$L_{G_{\mathrm{p}}} = \mathbb{E}_{x \sim p^{\pi_{\mathrm{p}}}, u \sim \pi_{\mathrm{p}}(\cdot|x), x' \sim P(\cdot|x,u), u' \sim \pi_{\mathrm{p}}(\cdot|x')} \left[ \left( G_{\mathrm{p}}(x, u) - (c(x) + (1 - c(x)) \gamma G_{\mathrm{p}}(x', u')) \right)^2 \right].$$

Among the four random variables involved in the expectation, $x$ and $u$ have shifted distributions because some of their samples are collected by the dual policy. To solve this problem, we use IS to correct their distribution. The IS ratio for $u$ can be readily computed by the ratio of probabilities under two policies. The IS ratio for $x$ involves the marginal state distribution under a policy, which can be computed as: $p^{\pi_{\mathrm{d}}}(x) = \sum_{t=0}^{\infty} P(x_t = x|\pi) = \sum_{t=0}^{\infty} \sum_{\tau} \mathbb{I}[x_t = x] p^{\pi_{\mathrm{d}}}(\tau)$, where $p^{\pi_{\mathrm{d}}}(\tau)$ is the probability of trajectory $\tau$ under $\pi_{\mathrm{d}}$:

$$p^{\pi_{\mathrm{d}}}(\tau) = p_{\mathrm{init}}(x_0) \pi_{\mathrm{d}}(u_0|x_0) P(x_1|x_0, u_0) \pi_{\mathrm{d}}(u_1|x_1) \cdots = p_{\mathrm{init}}(x_0) \prod_{t=0}^{\infty} \pi_{\mathrm{d}}(u_t|x_t) P(x_{t+1}|x_t, u_t).$$

To correct the distribution of $x$ from $p^{\pi_{\mathrm{d}}}$ to $p^{\pi_{\mathrm{p}}}$, we need to insert an IS ratio $r_{\mathrm{pd}}(x) = p^{\pi_{\mathrm{p}}}(x)/p^{\pi_{\mathrm{d}}}(x)$ into the loss function. However, directly computing the ratio is intractable because we cannot sum over all possible trajectories. Instead, we approximate the ratio with a single trajectory $\tau$ that contains $x$:

$$r_{\mathrm{pd}}(x) \approx \frac{p^{\pi_{\mathrm{p}}}(\tau)}{p^{\pi_{\mathrm{d}}}(\tau)} = \prod_{t=0}^{\infty} \frac{\pi_{\mathrm{p}}(u_t|x_t)}{\pi_{\mathrm{d}}(u_t|x_t)} \approx \prod_{s=0}^{t(x)} \frac{\pi_{\mathrm{p}}(u_s|x_s)}{\pi_{\mathrm{d}}(u_s|x_s)} := \hat{r}_{\mathrm{pd}}(x), \tag{8}$$

where the second approximation operation truncates the product up to $t(x)$, which is the appearing step of $x$. This is because future actions beyond $t(x)$ do not affect the probability of reaching $x$, and truncating them reduces the variance of IS. To additionally account for the distribution shift of $u$, we define the approximated IS ratio for a state-action pair as $\hat{r}_{\mathrm{pd}}(x, u) = \hat{r}_{\mathrm{pd}}(x) \pi_{\mathrm{p}}(u|x)/\pi_{\mathrm{d}}(u|x)$. The approximated IS ratios from the primal policy to the dual policy, $\hat{r}_{\mathrm{dp}}(x)$ and $\hat{r}_{\mathrm{dp}}(x, u)$, can be defined similarly.

The sequential multiplication in the IS ratio can easily cause numerical underflow since the probability of an action under another policy is usually lower than that under the behavior policy. We find that this problem can be effectively alleviated by constraining the KL divergence between the two policies. Observe that

$$D_{\mathrm{KL}}(\pi_{\mathrm{d}} \| \pi_{\mathrm{p}})[x] = \mathbb{E}_{u \sim \pi_{\mathrm{d}}(\cdot|x)} \left[ \log \frac{\pi_{\mathrm{d}}(u|x)}{\pi_{\mathrm{p}}(u|x)} \right] \leq \delta \iff \mathbb{E}_{u \sim \pi_{\mathrm{d}}(\cdot|x)} \left[ \log \frac{\pi_{\mathrm{p}}(u|x)}{\pi_{\mathrm{d}}(u|x)} \right] \geq -\delta.$$

The KL divergence constraint ensures that the expected logarithm of each term in the product (8) is not too small, thus preventing the IS ratio from collapsing to zero.

## 5.3 OVERALL ALGORITHM

Our algorithm, called feasible dual policy iteration (FDPI), follows the framework of feasible policy iteration (FPI) proposed by Yang et al. (2023c). On the basis of FPI, we combine the maximum entropy RL method from soft actor-critic (SAC) (Haarnoja et al., 2018) and name the resulting algorithm as SAC-FDPI. Our algorithm learns two action-feasibility networks $G_{\mathrm{p}, \phi_{\mathrm{p}}}, G_{\mathrm{d}, \phi_{\mathrm{d}}}$, an action-value network $Q_{\omega}$, and two policy networks $\pi_{\mathrm{p}, \theta_{\mathrm{p}}}, \pi_{\mathrm{d}, \theta_{\mathrm{d}}}$, where $\phi$, $\omega$, and $\theta$ denote their parameters. We additionally introduce a hyperparameter $\epsilon > 0$ and approximate feasibility by $G_{\phi}(x, u) \leq \epsilon$. This is because, in practice, approximation error causes the CDF to be positive almost everywhere since its learning target is non-negative. This approximation is valid under the assumption that the steps to violation is uniformly bounded (Thomas et al., 2021). In our experiments, we find that a fixed value of $\epsilon = 0.1$ works well for all environments.

The loss functions for the action-feasibility networks are

$$L_{G_{\#}}(\phi_{\#}) = \mathbb{E} \left[ \hat{r}_{\#}(x, u) \left( G_{\#, \phi_{\#}}(x, u) - y_{G_{\#}}(x, x', u') \right)^2 \right], \tag{9}$$

where "#" stands for "p" or "d", $\hat{r}_{\mathrm{p}}(x, u) = \hat{r}_{\mathrm{pd}}(x, u)$ for $(x, u)$ sampled by $\pi_{\mathrm{d}}$, and equals 1 for $(x, u)$ sampled by $\pi_{\mathrm{p}}$, $\hat{r}_{\mathrm{d}}(x, u)$ is defined similarly, and

$$y_{G_{\#}}(x, x', u') = c(x) + (1 - c(x)) \gamma G_{\#, \bar{\phi}_{\#}}(x', u'),$$

where $\bar{\phi}$ denote the parameters of the target networks. The loss functions for the action-value networks are

$$L_Q(\omega) = \mathbb{E}\left[\hat{r}_\mathrm{p}(x,u)\left(Q_\omega(x,u) - y_Q(x,x',u')\right)^2\right],$$

$$y_Q(x,x',u') = r(x,u) + \gamma(Q_{\bar{\omega}}(x',u') - \alpha \log \pi_\mathrm{p}(u'|x')), \quad (10)$$

where $\alpha$ is a learnable parameter for entropy temperature. The primal policy network is updated by maximizing the action-value network inside the feasible region and minimizing the primal action-feasibility network outside the feasible region. The loss function is as follows:

$$L_{\pi_\mathrm{p}}(\theta_\mathrm{p}) = \mathbb{E}\left[\hat{r}_\mathrm{p}(x)\left(\mathbb{I}_\mathrm{f}[x,u](\alpha \log \pi_{\mathrm{p},\theta_\mathrm{p}}(u|x) - Q_\omega(x,u)) + (1 - \mathbb{I}_\mathrm{f}[x,u])G_{\mathrm{p},\phi_\mathrm{p}}(x,u))\right)\right], \quad (11)$$

where $\mathbb{I}_\mathrm{f}[x,u] = 1$ if $G_{\mathrm{p},\phi_\mathrm{p}}(x,u) \leq \epsilon$ and 0 otherwise. The dual policy is updated by maximizing the dual action-feasibility network under the constraints of KL divergence. The constraints are addressed by the Lagrange multiplier method. The loss function for the dual policy is as follows:

$$L_{\pi_\mathrm{d}}(\theta_\mathrm{d}) = -\mathbb{E}\left[\hat{r}_\mathrm{d}(x)G_{\mathrm{d},\phi_\mathrm{d}}(x,u)\right] + \lambda_\mathrm{dp}D_\mathrm{KL}(\pi_{\mathrm{d},\theta_\mathrm{d}}\|\pi_{\mathrm{p},\theta_\mathrm{p}}) + \lambda_\mathrm{pd}D_\mathrm{KL}(\pi_{\mathrm{p},\theta_\mathrm{p}}\|\pi_{\mathrm{d},\theta_\mathrm{d}}). \quad (12)$$

The Lagrange multipliers $\lambda_\mathrm{dp}$ and $\lambda_\mathrm{dp}$ are updated by:

$$\lambda_\mathrm{dp} \leftarrow \left(\lambda_\mathrm{dp} + \eta\left(D_\mathrm{KL}(\pi_{\mathrm{d},\theta_\mathrm{d}}\|\pi_{\mathrm{p},\theta_\mathrm{p}}) - \delta\right)\right)_+, \quad \lambda_\mathrm{pd} \leftarrow \left(\lambda_\mathrm{pd} + \eta\left(D_\mathrm{KL}(\pi_{\mathrm{p},\theta_\mathrm{p}}\|\pi_{\mathrm{d},\theta_\mathrm{d}}) - \delta\right)\right)_+, \quad (13)$$

where $\eta$ is the learning rate and $(\cdot)_+$ denotes the projection to $\mathbb{R}_{\geq 0}$. The pseudocode of our algorithm is in Appendix B.

## 6 EXPERIMENTS

We aim to answer the following questions through our experiments:

**Q1** How does SAC-FDPI perform in terms of safety and return compared to existing algorithms?

**Q2** Does learning an additional dual policy help increase violating samples?

**Q3** Does the estimation error of the feasibility function decrease with more violating samples?

### 6.1 EXPERIMENT SETUPS

**Environments** Our experiments cover 14 environments in the Safety-Gymnasium benchmark (Ji et al., 2023a), including navigation and locomotion. The navigation environments include two robots, i.e., Point and Car, and four tasks, i.e., Goal, Push, Button, and Circle, with all difficulty levels set as 1 and constraints set as default. The locomotion environments include six classic robots from Gymnasium's MuJoCo environments, i.e., HalfCheetah, Hopper, Swimmer, Walker2d, Ant, and Humanoid, with maximum velocity constraints.

**Baselines** We compare our algorithm with a wide variety of mainstream safe RL algorithms implemented in the Omnisafe toolbox (Ji et al., 2023b), including iterative unconstrained RL algorithms RCPO (Tessler et al., 2018), PPO-Lag (Ray et al., 2019), and SAC-Lag (Ha et al., 2021), and constrained policy optimization algorithms CPO (Achiam et al., 2017), FOCOPS (Zhang et al., 2020), and CUP (Yang et al., 2022). In addition, we combine the state-of-the-art unconstrained RL algorithm DSAC-T (Duan et al., 2023) with the penalty method and name the resulting algorithm DSAC-T-Pen. We also include a version of our algorithm without the dual policy, named SAC-FPI. Hyperparameters for all algorithms are detailed in Appendix C.1.

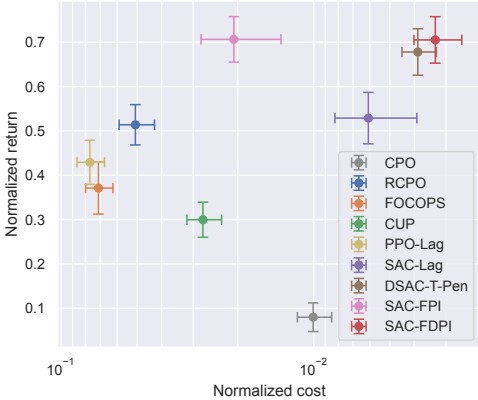

Figure 1: Normalized cost-return plot. Error bars represent 95% confidence intervals.

## 6.2 EXPERIMENT RESULTS

**Cost-return performance** In safe RL, we evaluate algorithms by two metrics: (1) episode cost, representing the average number of constraint-violating steps per episode, and (2) episode return, representing the average cumulative rewards per episode. To perform a comprehensive evaluation, we place the scores of all algorithms in a cost-return plot in Figure 1. The scores are first normalized by those of PPO and then averaged on all 14 environments. SAC-FDPI simultaneously achieves the lowest cost and an almost tied highest return (with SAC-FPI) among all algorithms, demonstrating its superior performance. We further plot the training curves of SAC-FDPI and six baselines across eight environments in Figure 2, with remaining results in Appendix C.2. SAC-FDPI achieves near-zero constraint violations on all environments while maintaining comparable or higher returns than the baselines. Notably, SAC-FPI exhibits persistent cost spikes even at final training stages, while SAC-FDPI maintains near-zero violations by continually feeding a controlled trickle of unsafe transitions through its dual policy. These results answer **Q1**.

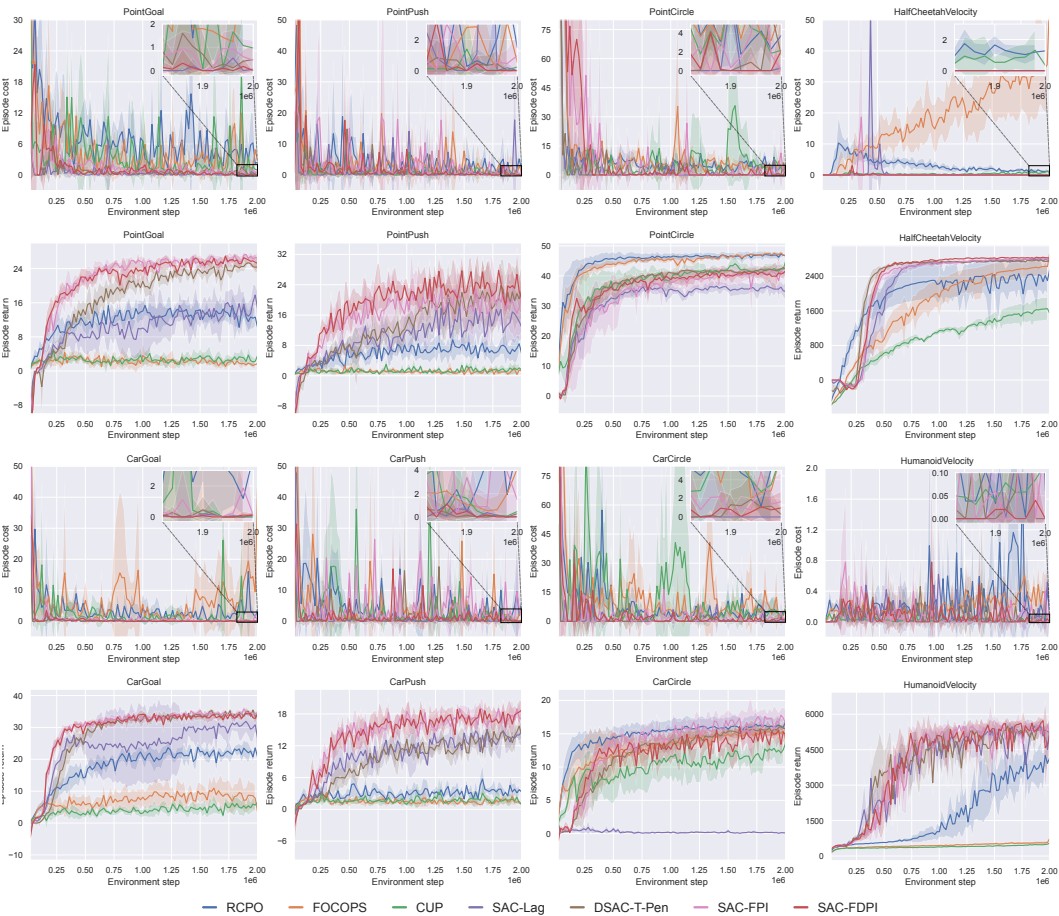

Figure 2: Training curves on eight environments in Safety-Gymnasium benchmark. The shaded areas represent 95% confidence intervals over 5 seeds.

**Proportion of violating samples** Figure 3 compares the average proportion of violating samples in the replay buffer during the final 10% of training iterations for SAC-FPI and SAC-FDPI. It shows that SAC-FDPI maintains a significantly higher proportion of violating samples than SAC-FPI—an order of magnitude greater in most environments. SAC-FPI's violation ratio falls below 1% in nearly all environments, undermining the accuracy of its feasibility function and leading to the cost spikes observed in its training curves. In contrast, SAC-FDPI's dual policy mechanism ensures persistent availability of a proper number of violating samples, which answers **Q2**. It is worth mentioning that our method is designed for training in a simulator instead of the real world, and thus additional violation does not cause real damage. A detailed discussion can be found in Appendix D.3.

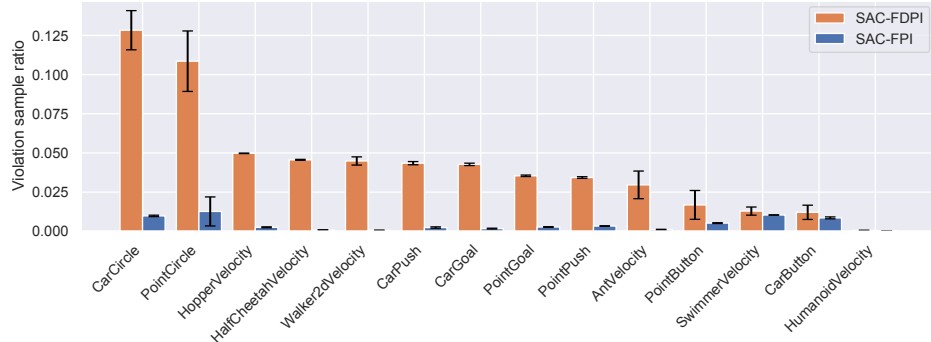

Figure 3: Average proportion of violating samples in the replay buffer. The error bars represent 95% confidence intervals over 5 seeds.

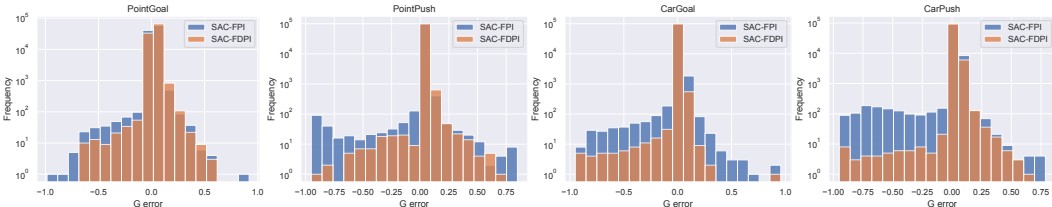

Figure 4: Distribution of feasibility function estimation error after convergence.

**Accuracy of feasibility function**    Figure 4 compares the estimation error of the feasibility function learned by SAC-FPI (blue) and SAC-FDPI (orange) in four environments. The errors are computed on 100k states collected by the two policies after convergence. The true values of the feasibility function are computed by definition on collected trajectories. It shows that SAC-FDPI produces a sharp, symmetric peak tightly centered at zero, which is an evidence of low bias and estimation error. In contrast, SAC-FPI exhibits flatter, more dispersed errors, reflecting the inflated estimation error bound caused by vanishing violating samples under the safety paradox. An empirical evidence of the relationship between violating samples and estimation error can be seen by combining Figures 3 and 4. Figure 3 shows that SAC-FDPI maintains about 10× more violating samples compared to SAC-FPI in most environments. This richer violation data directly leads to significantly lower estimation errors shown in Figure 4. These results support our theoretical analysis that richer violation data leads to better feasibility estimation, answering **Q3**.

**Exploration patterns**    To understand how the dual policy specifically helps collect violating samples, we visualize the trajectories of the primal policy and dual policy in Figure 5. The primal policy conservatively steers around the hazards so that no violation is incurred. In contrast, the dual policy augments the samples collected by the primal policy by deliberately cutting through the hazards, injecting constraint-violating samples while staying close to the primal policy. This richer mixture of safe and unsafe data keeps the feasibility function well-estimated, allowing the primal policy to converge to higher performance and with lower constraint violations.

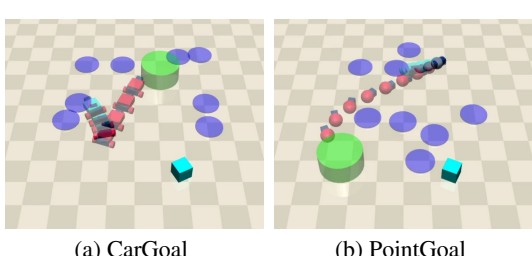

(a) CarGoal          (b) PointGoal

Figure 5: Trajectories of primal policy (red) and dual policy (cyan). Blue disks mark hazards, and green cylinder denotes the goal.

## 7 CONCLUSION

This paper discovers a fundamental obstacle in safe RL called the safety paradox, where improved policy safety leads to increased estimation error bound of the feasibility function, and ultimately harms policy safety. To address this paradox, we propose FDPI, which introduces a dual policy to maximize constraint violations while staying close to the primal policy through KL divergence constraints. We incorporate IS to correct distribution shifts between two policies. Extensive experiments on Safety-Gymnasium show that FDPI significantly increases violating samples, reduces feasibility function estimation error, and achieves state-of-the-art performance.

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

# A PROOFS

## A.1 PROOF OF CDF ERROR BOUND

**Theorem 1.** *For any infeasible state $x \in \mathcal{X}$ under policy $\pi$, let $\hat{F}^\pi(x)$ be the MC estimate of the CDF. Under Assumption 1, the expected relative estimation error is bounded by:*

$$\mathbb{E}_{\tau_1, \tau_2, \ldots, \tau_K} \left[ \left| \frac{\hat{F}^\pi(x) - F^\pi(x)}{F^\pi(x)} \right| \right] \leq \frac{1}{\sqrt{K}} |\ln \gamma| \sigma_N^\pi(x) + (\ln \gamma)^2 \frac{\sigma_N^{2,\pi}(x)}{\gamma^{\mu_N^\pi(x)}}. \tag{7}$$

*Proof.* Construct two auxiliary functions

$$H^\pi(x) = \gamma^{\mu_N^\pi(x)},$$
$$\hat{H}^\pi(x) = \gamma^{\hat{\mu}_N^\pi(x)},$$

where

$$\hat{\mu}_N^\pi(x) = \frac{1}{K} \sum_{i=1}^K N(\tau_i).$$

Perform a second-order Taylor expansion with the Lagrange remainder to $\hat{H}^\pi(x)$:

$$\hat{H}^\pi(x) = \gamma^{\mu_N^\pi(x)} + \gamma^{\mu_N^\pi(x)} \ln \gamma \cdot (\hat{\mu}_N^\pi(x) - \mu_N^\pi(x)) + \frac{1}{2} \gamma^M (\ln \gamma)^2 (\hat{\mu}_N^\pi(x) - \mu_N^\pi(x))^2,$$

where $M$ is a point between $\hat{\mu}_N^\pi(x)$ and $\mu_N^\pi(x)$. Thus,

$$|\hat{H}^\pi(x) - H^\pi(x)| = \left| \gamma^{\mu_N^\pi(x)} \ln \gamma \cdot (\hat{\mu}_N^\pi(x) - \mu_N^\pi(x)) + \frac{1}{2} \gamma^M (\ln \gamma)^2 (\hat{\mu}_N^\pi(x) - \mu_N^\pi(x))^2 \right|$$

$$\leq |\gamma^{\mu_N^\pi(x)} \ln \gamma \cdot (\hat{\mu}_N^\pi(x) - \mu_N^\pi(x))| + \frac{1}{2} |\gamma^M (\ln \gamma)^2 (\hat{\mu}_N^\pi(x) - \mu_N^\pi(x))^2|$$

$$\leq \gamma^{\mu_N^\pi(x)} |\ln \gamma| |\hat{\mu}_N^\pi(x) - \mu_N^\pi(x)| + \frac{1}{2} (\ln \gamma)^2 (\hat{\mu}_N^\pi(x) - \mu_N^\pi(x))^2,$$

where the last inequality holds because $\gamma^M < 1$. Since the squared function $(\cdot)^2$ is a convex function, by Jensen's inequality, we have

$$\mathbb{E}[|\hat{\mu}_N^\pi(x) - \mu_N^\pi(x)|] \leq \sqrt{\mathbb{E}[(\hat{\mu}_N^\pi(x) - \mu_N^\pi(x))^2]} = \frac{\sigma_N^\pi(x)}{\sqrt{K}}.$$

Thus,

$$\mathbb{E}[|\hat{H}^\pi(x) - H^\pi(x)|] \leq \gamma^{\mu_N^\pi(x)} |\ln \gamma| \mathbb{E}[|\hat{\mu}_N^\pi(x) - \mu_N^\pi(x)|] + \frac{1}{2} (\ln \gamma)^2 \mathbb{E}[(\hat{\mu}_N^\pi(x) - \mu_N^\pi(x))^2]$$

$$\leq \frac{1}{\sqrt{K}} |\ln \gamma| \gamma^{\mu_N^\pi(x)} \sigma_N^\pi(x) + \frac{1}{2K} (\ln \gamma)^2 \sigma_N^{2,\pi}(x). \tag{14}$$

For any trajectory $\tau_i$, perform a Taylor expansion to $\gamma^{N(\tau_i)}$:

$$\gamma^{N(\tau_i)} = \gamma^{\hat{\mu}_N^\pi(x)} + \gamma^{\hat{\mu}_N^\pi(x)} \ln \gamma \cdot (N(\tau_i) - \hat{\mu}_N^\pi(x)) + \frac{1}{2} \gamma^{M_i} (\ln \gamma)^2 (N(\tau_i) - \hat{\mu}_N^\pi(x))^2,$$

where $M_i$ is a point between $N(\tau_i)$ and $\hat{\mu}_N^\pi(x)$. Then,

$$|\hat{F}^\pi(x) - \hat{H}^\pi(x)| = \left| \frac{1}{K} \sum_{i=1}^K \gamma^{N(\tau_i)} - \gamma^{\hat{\mu}_N^\pi(x)} \right|$$

$$= \left| \gamma^{\hat{\mu}_N^\pi(x)} \ln \gamma \cdot \frac{1}{K} \sum_{i=1}^K (N(\tau_i) - \hat{\mu}_N^\pi(x)) + \frac{1}{2} (\ln \gamma)^2 \frac{1}{K} \sum_{i=1}^K \gamma^{M_i} (N(\tau_i) - \hat{\mu}_N^\pi(x))^2 \right|$$

$$= \frac{1}{2} (\ln \gamma)^2 \frac{1}{K} \sum_{i=1}^K \gamma^{M_i} (N(\tau_i) - \hat{\mu}_N^\pi(x))^2$$

$$\leq \frac{1}{2} (\ln \gamma)^2 \frac{1}{K} \sum_{i=1}^K (N(\tau_i) - \hat{\mu}_N^\pi(x))^2.$$

Using the relationship between sample variance and variance:

$$\mathbb{E}\left[\frac{1}{K-1}\sum_{i=1}^{K}(N(\tau_i)-\hat{\mu}_N^\pi(x))^2\right] = \sigma_N^{2,\pi}(x),$$

we have

$$\mathbb{E}[|\hat{F}^\pi(x)-\hat{H}^\pi(x)|] \le \frac{1}{2}(\ln\gamma)^2\mathbb{E}\left[\frac{1}{K}\sum_{i=1}^{K}(N(\tau_i)-\hat{\mu}_N^\pi(x))^2\right] \tag{15}$$

$$= \frac{K-1}{2K}(\ln\gamma)^2\sigma_N^{2,\pi}(x).$$

For any trajectory $\tau$, perform another Taylor expansion to $\gamma^{N(\tau_i)}$:

$$\gamma^{N(\tau_i)} = \gamma^{\mu_N^\pi(x)} + \gamma^{\mu_N^\pi(x)}\ln\gamma \cdot (N(\tau_i)-\mu_N^\pi(x)) + \frac{1}{2}\gamma^{M_i}(\ln\gamma)^2(N(\tau_i)-\mu_N^\pi(x))^2,$$

where $M_i$ is a point between $N(\tau_i)$ and $\mu_N^\pi(x)$. Then,

$$|F^\pi(x)-H^\pi(x)| = \left|\mathbb{E}[\gamma^{N(\tau_i)}]-\gamma^{\mu_N^\pi(x)}\right|$$

$$= \left|\gamma^{\mu_N^\pi(x)}\ln\gamma \cdot \mathbb{E}[N(\tau_i)-\mu_N^\pi(x)] + \frac{1}{2}(\ln\gamma)^2\mathbb{E}[\gamma^{M_i}(N(\tau_i)-\mu_N^\pi(x))^2]\right|$$

$$= \frac{1}{2}(\ln\gamma)^2\mathbb{E}[\gamma^{M_i}(N(\tau_i)-\mu_N^\pi(x))^2]$$

$$\le \frac{1}{2}(\ln\gamma)^2\mathbb{E}[(N(\tau_i)-\mu_N^\pi(x))^2]$$

$$= \frac{1}{2}(\ln\gamma)^2\sigma_N^{2,\pi}(x).$$

Thus,

$$\mathbb{E}[|F^\pi(x)-H^\pi(x)|] \le \frac{1}{2}(\ln\gamma)^2\sigma_N^{2,\pi}(x). \tag{16}$$

Combining (14), (15), and (16), we have

$$\mathbb{E}[|\hat{F}^\pi(x)-F^\pi(x)|] = \mathbb{E}[|\hat{F}^\pi(x)-\hat{H}^\pi(x)+\hat{H}^\pi(x)-H^\pi(x)+H^\pi(x)-F^\pi(x)|]$$

$$\le \mathbb{E}[|\hat{F}^\pi(x)-\hat{H}^\pi(x)|]+\mathbb{E}[|\hat{H}^\pi(x)-H^\pi(x)|]+\mathbb{E}[|H^\pi(x)-F^\pi(x)|]$$

$$\le \frac{1}{\sqrt{K}}|\ln\gamma|\gamma^{\mu_N^\pi(x)}\sigma_N^\pi(x) + \frac{1}{2K}(\ln\gamma)^2\sigma_N^{2,\pi}(x)$$

$$+ \frac{K-1}{2K}(\ln\gamma)^2\sigma_N^{2,\pi}(x) + \frac{1}{2}(\ln\gamma)^2\sigma_N^{2,\pi}(x)$$

$$= \frac{1}{\sqrt{K}}|\ln\gamma|\gamma^{\mu_N^\pi(x)}\sigma_N^\pi(x) + (\ln\gamma)^2\sigma_N^{2,\pi}(x)$$

Since $\gamma^N$ is a convex function for $\gamma \in (0,1)$, by Jensen's inequality, we have

$$F^\pi(x) = \mathbb{E}[\gamma^{N(\tau)}] \ge \gamma^{\mathbb{E}[N(\tau)]} = H^\pi(x).$$

Thus,

$$\mathbb{E}\left[\left|\frac{\hat{F}^\pi(x)-F^\pi(x)}{F^\pi(x)}\right|\right] \le \mathbb{E}\left[\left|\frac{\hat{F}^\pi(x)-F^\pi(x)}{H^\pi(x)}\right|\right] \le \frac{1}{\sqrt{K}}|\ln\gamma|\sigma_N^\pi(x) + (\ln\gamma)^2\frac{\sigma_N^{2,\pi}(x)}{\gamma^{\mu_N^\pi(x)}},$$

which proves the theorem. $\qquad\square$

### A.2 PROOF OF VARIANCE RELATIONSHIP

**Theorem 2.** *Let $\pi, \pi'$ be two policies. Consider a set of states $X_M \subseteq \mathcal{X}$ such that for all $x \in X_M$, (1) $x$ is infeasible under both $\pi$ and $\pi'$, (2) $m(x) \ge M$, (3) $\min\{P(D(x'_\pi) < D(x)), P(D(x'_{\pi'}) < D(x))\} \ge 0.5$. Under Assumptions 2–4, if $\pi'$ is safer than $\pi$ in all states in $X_M$, we have $\sigma_N^{2,\pi'}(x) \ge \sigma_N^{2,\pi}(x), \forall x \in X_M$.*

*Proof.* We prove the theorem in three steps.

**Step 1**: For a given policy, a state further to violation has a larger variance.

Let $x_+$ be a state with $D(x_+) \in [D(x), D(x) + \delta)$. To violate the constraint starting from $x_+$, the agent first needs move to some state $x_-$ with $D(x_-) \in [D(x) - \delta, D(x))$. Thus, we have

$$N^\pi(x_+) = N_1^\pi(x_+) + \sum_{i=2}^{m(x_+)} N_i^\pi(x_+) = N_1^\pi(x_+) + N^\pi(x_-).$$

Take the variance on both sides:

$$\sigma_N^{2;\pi}(x_+) = \text{Var}(N_1^\pi(x_+) + N^\pi(x_-))$$

$$= \text{Var}(N_1^\pi(x_+)) + \sigma_N^{2;\pi}(x_-) + 2\text{Cov}(N_1^\pi(x_+), N^\pi(x_-))$$

$$= \text{Var}(N_1^\pi(x_+)) + \sigma_N^{2;\pi}(x_-) + 2\text{Cov}\left(N_1^\pi(x_+), \sum_{i=2}^{m(x_+)} N_i^\pi(x_+)\right) \geq \sigma_N^{2;\pi}(x_-).$$

The last inequality follows from Assumption 4.

**Step 2**: In a given state, a safer policy has a larger variance.

Starting from $x$ and taking one step under $\pi$, the agent will arrive at a state $x_-$ with $D(x_-) \in [D(x) - \delta, D(x))$ with probability $P(D(x'_\pi) < D(x))$ and arrive at a state $x_+$ with $D(x_+) \in [(D(x), D(x) + \delta]$ with probability $1 - P(D(x'_\pi) < D(x))$. Let $p_\pi = P(D(x'_\pi) < D(x))$, we have

$$N^\pi(x) = \begin{cases} 1 + N^\pi(x_-) & \text{w.p. } p_\pi \\ 1 + N^\pi(x_+) & \text{w.p. } 1 - p_\pi \end{cases}$$

According to the law of total variance,

$$\sigma_N^{2;\pi}(x) = p_\pi \sigma_N^{2;\pi}(x_-) + (1 - p_\pi)\sigma_N^{2;\pi}(x_+) + p_\pi(1 - p_\pi)(\mu_N^\pi(x_-) - \mu_N^\pi(x_+))^2.$$

Consider a policy $\tilde{\pi}$ that is safer than $\pi$ in $x$ and identical to $\pi$ in all other states, i.e., $\tilde{\pi}$ only modifies the action in $x$ and follows $\pi$ thereafter. We have

$$p_{\tilde{\pi}} \leq p_\pi,$$
$$N^{\tilde{\pi}}(x_-) = N^\pi(x_-),$$
$$N^{\tilde{\pi}}(x_+) = N^\pi(x_+).$$

Since $\sigma_N^{2;\pi}(x_-) \leq \sigma_N^{2;\pi}(x_+)$, we have

$$p_{\tilde{\pi}}\sigma_N^{2;\tilde{\pi}}(x_-) + (1 - p_{\tilde{\pi}})\sigma_N^{2;\tilde{\pi}}(x_+) \geq p_\pi \sigma_N^{2;\pi}(x_-) + (1 - p_\pi)\sigma_N^{2;\pi}(x_+).$$

Since $p_\pi \geq p_{\tilde{\pi}} \geq 0.5$, we have

$$p_{\tilde{\pi}}(1 - p_{\tilde{\pi}})(\mu_N^{\tilde{\pi}}(x_-) - \mu_N^{\tilde{\pi}}(x_+))^2 \geq p_\pi(1 - p_\pi)(\mu_N^\pi(x_-) - \mu_N^\pi(x_+))^2.$$

Summing the above two inequalities, we have,

$$\sigma_N^{2;\tilde{\pi}}(x) \geq \sigma_N^{2;\pi}(x).$$

**Step 3**: For all states in $X_M$, a safer policy has a larger variance.

A safer policy can be obtained by modifying the actions state by state. Each time we modify the action in a single state, we obtain a safer policy in that state. The variance in that state increases, and the variance in other states remain unchanged. After modifying the actions in all states in $X_M$, we obtain the safer policy $\pi'$ with $\sigma_N^{2;\pi'}(x) \geq \sigma_N^{2;\pi}(x), \forall x \in X_M$, which proves the theorem.

Note that modifying the actions state by state is just a technique to facilitate the proof. It is not required in the actual policy update. The policy can be updated "globally" at once, which is a common practice for function approximated policies. The intermediate "virtual" policies, with each one safer in one state, are constructed only for the proof and do not need to be found in practice. $\square$

## A.3 CASE STUDY: ONE-DIMENSIONAL RANDOM WALK

Consider a one-dimensional random walk where the state space $\mathcal{X} = \mathbb{Z}$ and the action space $\mathcal{U} = \{-1, 1\}$. The initial state is fixed at $x = 0$, and the transition dynamics follows $x' = x + u$. In each step, the policy chooses $u = 1$ with probability $p$, and $u = -1$ with probability $1 - p$. We require $p > 0.5$ to ensure finite expectation and variance of steps to violation. The state $x = L > 0$ is constraint-violating, and every time the agent reaches this state, it is reset to $x = 0$. Let $N$ be the number of steps to the first violation, with its expectation and variance denoted as $\mu_N$ and $\sigma_N^2$. To measure policy safety, let $r$ be the expected proportion of violating samples under continuous sampling. It is easily observed that $r = 1/\mu_N$. Our aim is to find the relationship between $r$ and $\sigma_N^2$.

Observe that $N$ can be decomposed as the sum of $L$ random variables: $N = LM$, where $M$ is the number of steps to the first time the agent visits the step on its right. This is because going from $x = 0$ to $x = L$ can be decomposed as going to the step on the right $L$ times. We derive the expectation and variance of $M$, denoted as $\mu_M$ and $\sigma_M^2$, respectively. Consider the following two cases:

1. With probability $p$, the agent moves right and reaches $x + 1$ in one step.
2. With probability $1 - p$, the agent moves left to $x - 1$. Now, the agent must first return to $x$ and then proceed to $x + 1$. The expected number of steps of returning to $x$ from $x - 1$ is the same as that of moving from $x$ to $x + 1$, which equals $\mu_M$. Thus, the total expected number of steps in the case is $1 + 2\mu_M$.

Thus, the expectation $\mu_M$ satisfies

$$\mu_M = p \cdot 1 + (1 - p) \cdot (1 + 2\mu_M),$$

which gives

$$\mu_M = \frac{1}{2p - 1}.$$

To derive the variance of $M$, first consider the second-order moment $\mathbb{E}[M^2]$. Following the above analysis, we have

$$\mathbb{E}[M^2] = p \cdot 1^2 + (1 - p) \cdot \mathbb{E}[(1 + M' + M)^2],$$

where $M'$ is the number of steps to move from $x - 1$ to $x$ for the first time. Since $M'$ and $M$ are i.i.d., we have

$$
\begin{aligned}
\mathbb{E}[(1 + M' + M)^2] &= \mathbb{E}[1 + 2(M' + M) + (M' + M)^2] \\
&= 1 + 4\mu_M + \mathbb{E}[(M' + M)^2] \\
&= 1 + 4\mu_M + \mathbb{E}[M'^2 + 2M'M + M^2] \\
&= 1 + 4\mu_M + 2\mathbb{E}[M^2] + 2\mu_M^2.
\end{aligned}
$$

Thus,

$$\mathbb{E}[M^2] = p + (1 - p) \cdot (1 + 4\mu_M + 2\mathbb{E}[M^2] + 2\mu_M^2).$$

Solving for $\mathbb{E}[M^2]$:

$$\mathbb{E}[M^2] = \frac{1 + 4(1 - p)\mu_M + 2(1 - p)\mu_M^2}{2p - 1}.$$

Thus, the variance is

$$\sigma_M^2 = \mathbb{E}[M^2] - \mu_M^2 = \frac{4p(1 - p)}{(2p - 1)^3}.$$

Since $N = LM$, by the property of the summation of random variables, we have

$$\mu_N = L\mu_M = \frac{1}{2p - 1} \cdot L,$$

$$\sigma_N^2 = L^2\sigma_M^2 = \frac{4p(1 - p)}{(2p - 1)^3} \cdot L^2.$$

With $r = 1/\mu_N$, we have

$$L = (2p - 1)\mu_N = \frac{2p - 1}{r}.$$

Thus,

$$\sigma_N^2 = \frac{4p(1-p)}{(2p-1)^3} \cdot \frac{(2p-1)^2}{r^2} = \frac{4p(1-p)}{2p-1} \cdot \frac{1}{r^2}$$

The above equation reveals that as the policy becomes safer, the variance of steps to violation increases. Combining with the error bound in Equation (7), we conclude that a safer policy leads to a higher relative estimation error bound.

## B PSEUDOCODE

---

**Algorithm 1:** Soft actor-critic with feasible dual policy iteration (SAC-FDPI)

---

**Initialize:** Network parameters $\phi_p$, $\phi_d$, $\omega$, $\theta_p$, $\theta_d$. IS ratios $\hat{r}_{dp} = \hat{r}_{pd} = 1$. Replay buffer $\mathcal{D} = \emptyset$.
Feasibility threshold $d = 0.95$.

1 **for** *each iteration* **do**
    // Sample with primal policy
2    Sample action $u_p \sim \pi_{\theta_p}$;
3    Get next state $x'$, reward $r$, and cost $c$ from environment;
4    Store transition in replay buffer $\mathcal{D} \leftarrow \mathcal{D} \cup \{(x, u, x', r, c, \hat{r}_{dp})\}$;
5    Update IS ratio $\hat{r}_{dp} \leftarrow \hat{r}_{dp} \cdot \pi_d(u|x)/\pi_p(u|x)$ or $\hat{r}_{dp} \leftarrow 1$ if episode ends;
6    **if** *Running averaged proportion of $G_p(x, u) \leq \epsilon$ greater than $d$* **then**
        // Sample with dual policy
7        Sample action $\tau_d \sim \pi_{\theta_d}$;
8        Get next state $x'$, reward $r$, and cost $c$ from environment;
9        Store transition in replay buffer $\mathcal{D} \leftarrow \mathcal{D} \cup \{(x, u, x', r, c, \hat{r}_{pd})\}$;
10       Update IS ratio $\hat{r}_{pd} \leftarrow \hat{r}_{pd} \cdot \pi_p(u|x)/\pi_d(u|x)$ or $\hat{r}_{pd} \leftarrow 1$ if episode ends;
11    **else**
        // Sample with primal policy
12       Repeat Lines 2–5;
13    **end**
    // Update network parameters
14    Update primal action-feasibility network $\phi_p \leftarrow \phi_p - \eta \nabla_{\phi_p} L_{G_p}(\phi_p)$; // Equation (9)
15    Update dual action-feasibility network $\phi_d \leftarrow \phi_d - \eta \nabla_{\phi_d} L_{G_d}(\phi_d)$;  // Equation (9)
16    Update action-value network $\omega \leftarrow \omega - \eta \nabla_\omega L_Q(\omega)$;            // Equation (10)
17    Update primal policy network $\theta_p \leftarrow \theta_p - \eta \nabla_{\theta_p} L_{\pi_p}(\theta_p)$;      // Equation (11)
18    Update dual policy network $\theta_d \leftarrow \theta_d - \eta \nabla_{\theta_d} L_{\pi_d}(\theta_d)$;       // Equation (12)
19    Update Lagrange multipliers by Equation (13);
20 **end**

---

# C  EXPERIMENTS

The Safety-Gymnasium benchmark (Ji et al., 2023a) and the Omnisafe toolbox (Ji et al., 2023b) are both released under the Apache License 2.0.

All experiments are conducted on a workstation equipped with Intel(R) Xeon(R) Gold 6246R CPUs (32 cores, 64 threads), an NVIDIA GeForce RTX 3090 GPU, and 256GB of RAM. A single experimental trial—comprising one environment, one algorithm, and one random seed—takes about 2 hours to execute. Executing all experiments with a properly configured concurrent running scheme requires approximately 400 hours.

## C.1  HYPERPARAMETERS

Table 1: Hyperparameters

| Category | Hyperparameter | Value |
|---|---|---|
| Shared | Number of vector environments | 2 |
| | Number of samples per iteration | 2 |
| | Number of updates per iteration | 1 |
| | Replay buffer size | 2e6 |
| | Batch size | 256 |
| | Reward discount factor | 0.99 |
| | Cost discount factor | 0.97 |
| | Cost limit | 0 |
| | Actor learning rate | 1e-4 |
| | Actor network hidden sizes | (256, 256) |
| | Actor activation function | ReLU |
| | Critic learning rate | 1e-4 |
| | Critic network hidden sizes | (256, 256) |
| | Critic activation function | ReLU |
| | Network weight initialization method | Truncated normal |
| | Optimizer | Adam |
| | Target network soft update weight | 0.005 |
| | Maximum gradient norm | 40 |
| SAC | Initial entropy temperature | 1.0 |
| | Target entropy | $-\dim(\mathcal{U})$ |
| | Entropy temperature learning rate | 1e-4 |
| Penalty | Penalty coefficient | 1.0 |
| Lagrangian | Initial multiplier | 0.0 |
| | Multiplier learning rate | 1e-4 |
| | Multiplier update delay | 10 |
| FDPI | Primal policy step per iteration | 1 |
| | Dual policy step per iteration | 1 |
| | Feasibility threshold $\epsilon$ | 0.1 |
| | Maximum KL divergence $\delta$ | 5.0 |

## C.2  ADDITIONAL RESULTS

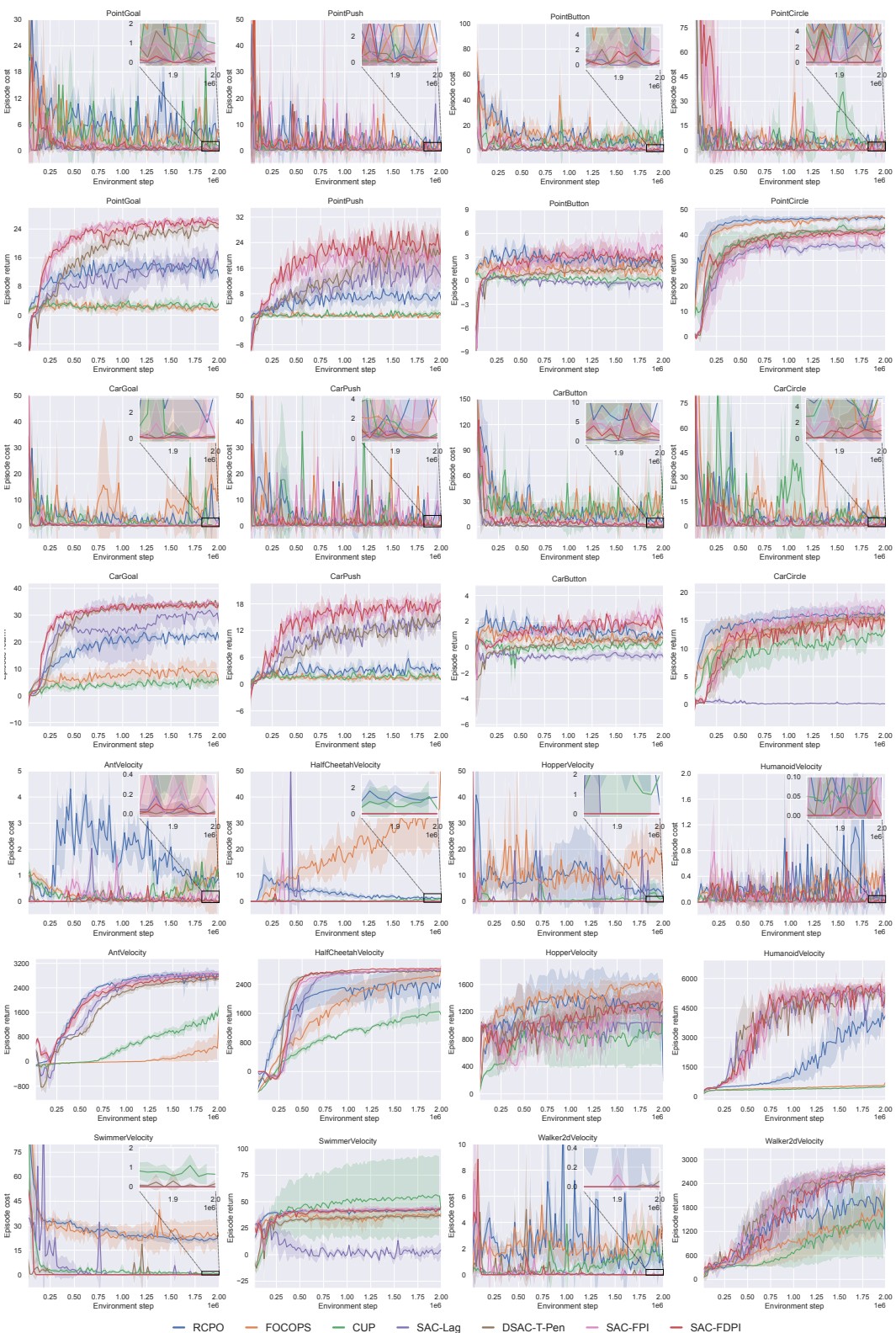

Figure 6: Training curves on all 14 environments in Safety-Gymnasium benchmark. The shaded areas represent 95% confidence intervals over 5 seeds.

Table 2: Average cost and return in the last 10% iterations. Mean $\pm$ Std over 5 seeds.

| Algorithm | PointGoal | | PointPush | | HalfCheetahVelocity | |
|---|---|---|---|---|---|---|
| | Cost | Return | Cost | Return | Cost | Return |
| CPO | $0.90 \pm 0.45$ | $1.45 \pm 0.99$ | $0.60 \pm 0.66$ | $0.49 \pm 0.75$ | $0.03 \pm 0.02$ | $855.39 \pm 239.61$ |
| RCPO | $4.08 \pm 1.18$ | $13.14 \pm 1.92$ | $1.96 \pm 0.81$ | $7.24 \pm 1.10$ | $1.29 \pm 0.38$ | $2389.86 \pm 341.21$ |
| FOCOPS | $3.16 \pm 0.83$ | $1.85 \pm 0.56$ | $2.21 \pm 1.18$ | $1.14 \pm 0.28$ | $34.15 \pm 14.09$ | $2606.12 \pm 214.08$ |
| CUP | $3.01 \pm 3.24$ | $2.91 \pm 0.98$ | $0.91 \pm 0.51$ | $1.37 \pm 0.52$ | $0.91 \pm 0.90$ | $1619.77 \pm 259.74$ |
| PPO-Lag | $5.55 \pm 3.37$ | $3.83 \pm 2.13$ | $5.17 \pm 3.86$ | $1.89 \pm 1.23$ | $2.67 \pm 1.54$ | $2234.27 \pm 499.64$ |
| SAC-Lag | $\mathbf{0.09} \pm 0.10$ | $14.66 \pm 3.01$ | $2.27 \pm 3.29$ | $14.48 \pm 3.38$ | $\mathbf{0.00} \pm 0.00$ | $2783.54 \pm 26.26$ |
| DSAC-T-Pen | $0.58 \pm 0.28$ | $24.65 \pm 0.62$ | $\mathbf{0.38} \pm 0.27$ | $\mathbf{21.13} \pm 6.82$ | $\mathbf{0.00} \pm 0.00$ | $2760.54 \pm 14.76$ |
| SAC-FPI | $0.61 \pm 0.24$ | $\mathbf{26.45} \pm 0.41$ | $0.78 \pm 0.52$ | $19.05 \pm 6.63$ | $\mathbf{0.00} \pm 0.00$ | $\mathbf{2809.33} \pm 23.42$ |
| SAC-FDPI | $\mathbf{0.09} \pm 0.06$ | $\mathbf{25.77} \pm 0.49$ | $\mathbf{0.44} \pm 0.67$ | $\mathbf{22.71} \pm 1.21$ | $\mathbf{0.00} \pm 0.00$ | $\mathbf{2831.97} \pm 9.13$ |

| Algorithm | Walker2dVelocity | | HumanoidVelocity | | SafetyHopper | |
|---|---|---|---|---|---|---|
| | Cost | Return | Cost | Return | Cost | Return |
| CPO | $0.02 \pm 0.01$ | $188.35 \pm 96.17$ | $\mathbf{0.00} \pm 0.00$ | $250.84 \pm 18.05$ | $519.61 \pm 509.11$ | $-4599.10 \pm 6207.66$ |
| RCPO | $1.11 \pm 0.54$ | $1913.68 \pm 765.27$ | $0.28 \pm 0.10$ | $3862.34 \pm 405.31$ | $577.12 \pm 413.10$ | $1481.77 \pm 639.77$ |
| FOCOPS | $2.72 \pm 0.99$ | $1629.19 \pm 516.49$ | $0.35 \pm 0.13$ | $562.29 \pm 74.98$ | $763.58 \pm 19.10$ | $1332.04 \pm 297.91$ |
| CUP | $1.60 \pm 0.59$ | $1360.46 \pm 814.21$ | $0.06 \pm 0.03$ | $482.02 \pm 34.07$ | $206.51 \pm 213.09$ | $-1266.47 \pm 1504.74$ |
| PPO-Lag | $2.00 \pm 0.61$ | $2166.22 \pm 595.72$ | $0.22 \pm 0.07$ | $1300.45 \pm 425.44$ | $507.99 \pm 209.61$ | $1337.05 \pm 204.90$ |
| SAC-Lag | $\mathbf{0.00} \pm 0.00$ | $2681.36 \pm 126.54$ | $0.10 \pm 0.09$ | $5212.90 \pm 110.20$ | $84.39 \pm 20.99$ | $1585.24 \pm 1252.75$ |
| DSAC-T-Pen | $0.01 \pm 0.02$ | $\mathbf{2752.25} \pm 106.80$ | $0.05 \pm 0.05$ | $5229.07 \pm 176.58$ | $6.70 \pm 4.83$ | $\mathbf{3122.00} \pm 179.75$ |
| SAC-FPI | $0.01 \pm 0.02$ | $\mathbf{2787.28} \pm 126.27$ | $0.09 \pm 0.08$ | $\mathbf{5351.34} \pm 183.96$ | $\mathbf{2.23} \pm 3.15$ | $2947.83 \pm 227.69$ |
| SAC-FDPI | $\mathbf{0.00} \pm 0.00$ | $2619.63 \pm 148.14$ | $\mathbf{0.01} \pm 0.01$ | $\mathbf{5269.77} \pm 201.84$ | $\mathbf{0.33} \pm 0.39$ | $\mathbf{3118.77} \pm 75.85$ |

*Note:* Bold values indicate top 2 algorithms. Colored cells indicate top 2 in both cost and return.

We evaluated the feasibility threshold $\epsilon \in \{0.05, 0.1, 0.2\}$ across 8 environments. The results indicate that a smaller $\epsilon$ leads to more conservative behavior, i.e., lower cost and lower return.

Table 3: Normalized cost and return under different feasibility threshold $\epsilon$

| Environment | $\epsilon = 0.05$ | | $\epsilon = 0.1$ | | $\epsilon = 0.2$ | |
|---|---|---|---|---|---|---|
| | Cost | Return | Cost | Return | Cost | Return |
| PointGoal | 0.001 | 0.998 | 0.002 | 1.073 | 0.017 | 1.080 |
| PointPush | 0.000 | 0.822 | 0.002 | 0.770 | 0.001 | 0.816 |
| PointCircle | 0.000 | 0.703 | 0.004 | 0.797 | 0.040 | 0.862 |
| CarGoal | 0.000 | 1.087 | 0.001 | 1.115 | 0.000 | 1.085 |
| CarPush | 0.001 | 0.862 | 0.016 | 0.934 | 0.008 | 0.888 |
| CarCircle | 0.001 | 0.681 | 0.003 | 0.619 | 0.016 | 0.824 |
| HalfCheetahVelocity | 0.000 | 1.386 | 0.000 | 1.385 | 0.000 | 1.418 |
| HumanoidVelocity | 0.001 | 0.487 | 0.000 | 0.810 | 0.000 | 0.736 |
| Average | 0.000 | 0.878 | 0.003 | 0.938 | 0.010 | 0.964 |

We evaluated the dual threshold $d \in \{0.5, 0.9, 0.95, 0.98\}$. The results show that within a reasonable range (i.e., for $d \geq 0.9$), a smaller $d$, which corresponds to more frequent activation of the dual policy, leads to lower costs without sacrificing return. However, an excessively small $d$ results in higher costs, possibly because of severe distributional shift.

Table 4: Normalized cost and return under different dual threshold $d$

| Environment | $d = 0.50$ | | $d = 0.90$ | | $d = 0.95$ | | $d = 0.98$ | |
|---|---|---|---|---|---|---|---|---|
| | Cost | Return | Cost | Return | Cost | Return | Cost | Return |
| PointGoal | 0.001 | 0.988 | 0.000 | 1.091 | 0.002 | 1.073 | 0.008 | 1.076 |
| PointPush | 0.003 | 0.747 | 0.000 | 0.746 | 0.002 | 0.770 | 0.005 | 0.798 |
| PointCircle | 0.004 | 0.773 | 0.000 | 0.787 | 0.004 | 0.797 | 0.019 | 0.861 |
| CarGoal | 0.003 | 1.107 | 0.000 | 1.142 | 0.001 | 1.115 | 0.000 | 1.110 |
| CarPush | 0.150 | 0.943 | 0.005 | 1.028 | 0.016 | 0.934 | 0.021 | 0.789 |
| CarCircle | 0.000 | 0.625 | 0.001 | 0.739 | 0.003 | 0.619 | 0.001 | 0.703 |
| HalfCheetahVelocity | 0.000 | 1.374 | 0.000 | 1.382 | 0.000 | 1.385 | 0.000 | 1.374 |
| HumanoidVelocity | 0.000 | 0.825 | 0.000 | 0.682 | 0.000 | 0.810 | 0.000 | 0.763 |
| Average | 0.020 | 0.923 | 0.001 | 0.949 | 0.003 | 0.938 | 0.007 | 0.934 |

We evaluated the KL divergence threshold $\delta \in \{2, 5, 10\}$. The results show that the overall performance is stable within a reasonable range ($\delta \leq 5$), with a smaller $\delta$ slightly decreases both cost and return. However, an excessively large $\delta$ significantly increases cost due to excessive policy divergence.

Table 5: Normalized cost and return under different KL divergence threshold $\delta$

| Environment | $\delta = 2$ | | $\delta = 5$ | | $\delta = 10$ | |
|---|---|---|---|---|---|---|
| | Cost | Return | Cost | Return | Cost | Return |
| PointGoal | 0.002 | 1.000 | 0.002 | 1.073 | 0.015 | 1.036 |
| PointPush | 0.000 | 0.335 | 0.002 | 0.770 | 0.061 | 0.647 |
| PointCircle | 0.005 | 0.775 | 0.004 | 0.797 | 0.012 | 0.861 |
| CarGoal | 0.000 | 1.108 | 0.001 | 1.115 | 0.002 | 1.144 |
| CarPush | 0.010 | 0.857 | 0.016 | 0.934 | 0.387 | 0.021 |
| CarCircle | 0.002 | 0.766 | 0.003 | 0.619 | 0.005 | 0.824 |
| HalfCheetahVelocity | 0.000 | 1.433 | 0.000 | 1.385 | 0.000 | 1.425 |
| HumanoidVelocity | 0.000 | 0.843 | 0.000 | 0.810 | 0.000 | 0.709 |
| Average | 0.002 | 0.890 | 0.003 | 0.938 | 0.060 | 0.833 |

We compared SAC-FDPI with IPO (Liu et al., 2020) and CRPO (Xu et al., 2021) on 8 environments. The results show that SAC-FDPI achieves a lower cost and higher return than these two algorithms.

Table 6: Normalized cost and return comparison with IPO and CRPO.

| Environment | IPO | | CRPO | | SAC-FDPI | |
|---|---|---|---|---|---|---|
| | Cost | Return | Cost | Return | Cost | Return |
| PointGoal | 0.060 | 0.086 | 0.142 | 0.607 | 0.002 | 1.073 |
| PointPush | 0.188 | 0.063 | 0.057 | 0.193 | 0.002 | 0.770 |
| PointCircle | 0.052 | 0.816 | 0.013 | 0.918 | 0.004 | 0.797 |
| CarGoal | 0.122 | 0.555 | 0.123 | 0.546 | 0.001 | 1.115 |
| CarPush | 0.551 | 0.131 | 0.831 | 0.175 | 0.016 | 0.934 |
| CarCircle | 0.045 | 0.624 | 0.025 | 0.677 | 0.003 | 0.619 |
| HalfCheetahVelocity | 0.073 | 1.435 | 0.004 | 1.146 | 0.000 | 1.385 |
| HumanoidVelocity | 0.001 | 0.162 | 0.002 | 0.635 | 0.000 | 0.810 |
| Average | 0.136 | 0.484 | 0.149 | 0.612 | 0.003 | 0.938 |

# D DISCUSSION

## D.1 RELATED WORK ON REPRESENTATION LEARNING

Another class of methods address the challenge of sparse violation signals in feasibility function estimation through representation learning. Cen et al. (2024) propose Feasibility Consistent Safe Reinforcement Learning (FCSRL), which extracts safety-related information from the observation to improve feasibility function learning. This is achieved by learning a encoder that maps the observation to a latent states, which serves as a better input for the feasibility function. FCSRL focuses on representation learning, which tries to better exploit safety-related information from given data. In contrast, our method addresses the safety paradox, where violation signals become increasingly sparse as the policy becomes safer. Our solution is to actively collect more violating data using a dual policy, targeting the root cause of the sparsity. Therefore, FCSRL and our method are orthogonal: one improves data utilization, while the other improves data collection.

## D.2 EXTENSION TO TD ESTIMATE AND CVF

Our theoretical analysis can be extended to TD estimate. Specifically, Section 4.2 proves that the variance of steps to violation, $N(\tau)$, increases with safer policies. Consider an initial state $x$, its subsequent state $x'$, and the sub-trajectory $\tau'$ starting from $x'$. Since $N(\tau') = N(\tau) - 1$, the variance of $N(\tau')$ also increases with policy safety. This implies a higher variance in the true feasibility value $F^\pi(x')$. Consequently, the TD target, which is computed by $F^\pi(x')$, inherits this increased variance, leading to a larger estimation error in $\hat{F}^\pi(x)$.

Our analysis can also be extended to the CVF widely used in the CMDP, defined as

$$F^\pi(x) = \mathbb{E}_{\tau \sim \pi} \left[ \sum_{t=0}^{\infty} \gamma^t c(x_t) | x_0 = x \right].$$

The key insight is that the CVF can be decomposed into a discounted sum of CDF-like terms. Specifically, we can break down any infinite-horizon trajectory into segments that end immediately after a constraint violation. The total CVF is the discounted sum of costs along the entire trajectory. We can now group the costs by the segment in which they occur. The cost incurred in each segment is, by construction, a discounted sum that starts from an initial state and ends with a violation. Crucially, the value of the CVF in each segment is precisely a CDF. The total CVF can be expressed as:

$$F^\pi(x) = \underbrace{\mathbb{E}[\text{discounted cost of Segment 1}]}_{\text{CDF term}} + \gamma^{T_1} \underbrace{\mathbb{E}[\text{discounted cost of Segment 2}]}_{\text{CDF term}} + \cdots$$

where $T_i$ is the time step of the $i$th violation. Our main theoretical result establishes that the estimation error bound of a single CDF term increases as the policy becomes safer (i.e., as violations become rarer, making each segment longer). This directly implies that the error bound for the total CVF must also increase.

## D.3 ADDITIONAL VIOLATION DURING TRAINING

In the broader safe RL community, there are two training and implementation modes: (1) offline training and online deployment (OTOD), which first trains a policy in simulator and then deploys it in the real world, and (2) simultaneous online training and deployment (SOTD), which directly interacts with the real world to collect data for training. The OTOD mode only requires the final policy to be safe because intermediate policies will not be deployed in the real world. The SOTD model requires both the final policy and all intermediate policies to be safe.

FDPI, along with the baselines we compare in the paper, belongs to the OTOD mode. We focus primarily on learning a safe policy at convergence, rather than guaranteeing safety during training. Therefore, FDPI does not involve unsafe exploration in the real world during training.

For the SOTD mode, one must ensure safety throughout training. Achieving this goal would require integrating additional safe exploration techniques like those proposed by Berkenkamp et al. (2017) and Yu et al. (2022). These methods typically employ a model of the environment, either known or

learned. Constraint violations are allowed in the model but not allowed in the environment. They alternate between learning a safe policy within the current model and refining the model with newly collected data. These approaches are complementary to our contribution, which focuses on solving the safe policy within a fixed model/environment.

## E   LARGE LANGUAGE MODEL USAGE DISCLOSURE

We used Large Language Model (LLM) solely for the purpose of improving grammar and polishing writing. The LLM was not used for any core research tasks such as retrieval, discovery, ideation, or analysis.

