# OpenReview forum: "Breaking Safety Paradox with Feasible Dual Policy Iteration"
_ICLR.cc/2026/Conference — ICLR 2026 Poster_

### Official Review · Reviewer_W7n2 · 2025-10-28

**Soundness:** 3
**Presentation:** 3
**Contribution:** 2
**Rating:** 6
**Confidence:** 3

**Summary:**

**DISCLAIMER: I was one of the reviewers of this paper at an earlier conference.**

The paper proposes "safety paradox" as a main obstacle towards zero constraint violation in safe RL, where the improvement in policy safety causes the estimation error of the feasibility function to increase and in turn harms policy safety. An algorithm is proposed to solve this issue. It works through introducing another policy that seeks to violate the constraint as much as possible, so as to provide more samples for the training of the feasibility function. Importance sampling and KL-divergence constraints are employed to mitigate the distributional shifts between the this newly introduced policy and the original learning policy. Empirical valuation shows the effectiveness of the proposed method.

**Strengths:**

- The flow of this paper is clear and easy to follow.
- Achieving zero constraint violations has great practical significance, and the estimation error of the feasibility function has been a key issue to this. In this regard, this work is well-motivated.
- I did not go through the proof, but the theoretical analysis makes sense to me, and can serve as a valid justification of the proposed safety paradox.
- The experiment results of the proposed method look quite promising.

**Weaknesses:**

- The theory (Theorem 1) only shows that the upper bound of the estimation error will become larger, not exactly the error itself. I know showing how the exact error relates to constraint violations may be too much to ask, so I recommend showing some empirical evidence about this, which would be a good supplement for the theory. (In the previous rebuttal phase, the authors gave empirical evidence demonstrating the relationship between constraint-violating samples and estimation error. I would appreciate it if the results can be added to the paper.)
- The proposed method solves the safety paradox by introducing another policy that aims to intentionally violate the constraint, which causes extra cost during training and thus may not be practical in the real world.
- There are some other existing baselines this paper is missing: IPO [1] and CRPO [2]. In the previous rebuttal phase, the authors showed their performance. I would appreciate it if the results can be added to the paper.
- The metric used in the empirical evaluation may be a problem. See Questions.

[1]: Liu et al., 2020, Ipo: Interior-point policy optimization under constraints.

[2]: Xu et al., 2021, Crpo: A new approach for safe reinforcement learning with convergence guarantee.

**Questions:**

My questions were resolved during the previous rebuttal phase of this paper, at an earlier conference. The only thing left: It would be nice if the additional results and discussions from the previous rebuttal can be added to the paper.

---

> ### Author Response · Authors · 2025-11-23
> **Authors' Response**
>
> We sincerely thank the reviewer for their continued engagement with our work and for providing valuable feedback throughout the review process. We truly appreciate the time and effort invested in helping us improve the paper. We have uploaded a revised version of the paper, where all the results and discussions are included.
>
> ## Weaknesses
>
> > The theory (Theorem 1) only shows that the upper bound of the estimation error will become larger, not exactly the error itself. I know showing how the exact error relates to constraint violations may be too much to ask, so I recommend showing some empirical evidence about this, which would be a good supplement for the theory. (In the previous rebuttal phase, the authors gave empirical evidence demonstrating the relationship between constraint-violating samples and estimation error. I would appreciate it if the results can be added to the paper.)
>
> We have added an explanation after Fig. 4 in Section 6.2:
>
> "An empirical evidence of the relationship between violating samples and estimation error can be seen by combining Figures 3 and 4. Figure 3 shows that SAC-FDPI maintains about 10× more violating samples compared to SAC-FPI in most environments. This richer violation data directly leads to significantly lower estimation errors shown in Figure 4. These results support our theoretical analysis that richer violation data leads to better feasibility estimation."
>
> > The proposed method solves the safety paradox by introducing another policy that aims to intentionally violate the constraint, which causes extra cost during training and thus may not be practical in the real world.
>
> We have added an explanation after Fig. 3 in Section 6.2 and in Appendix D.3:
>
> "It is worth mentioning that our method is designed for training in a simulator instead of the real world, and thus additional violation does not cause real damage. A detailed discussion can be found in Appendix D.3."
>
> "In the broader safe RL community, there are two training and implementation modes: (1) offline training and online deployment (OTOD), which first trains a policy in simulator and then deploys it in the real world, and (2) simultaneous online training and deployment (SOTD), which directly interacts with the real world to collect data for training. The OTOD mode only requires the final policy to be safe because intermediate policies will not be deployed in the real world. The SOTD model requires both the final policy and all intermediate policies to be safe."
>
> "FDPI, along with the baselines we compare in the paper, belongs to the OTOD mode. We focus primarily on learning a safe policy at convergence, rather than guaranteeing safety during training. Therefore, FDPI does not involve unsafe exploration in the real world during training."
>
> "For the SOTD mode, one must ensure safety throughout training. Achieving this goal would require integrating additional safe exploration techniques like those proposed by Berkenkamp et al. (2017) and Yu et al. (2022). These methods typically employ a model of the environment, either known or learned. Constraint violations are allowed in the model but not allowed in the environment. They alternate between learning a safe policy within the current model and refining the model with newly collected data. These approaches are complementary to our contribution, which focuses on solving the safe policy within a fixed model/environment."
>
> > There are some other existing baselines this paper is missing: IPO [1] and CRPO [2]. In the previous rebuttal phase, the authors showed their performance. I would appreciate it if the results can be added to the paper.
> >
> > [1] Liu et al., 2020, Ipo: Interior-point policy optimization under constraints.
> >
> > [2] Xu et al., 2021, Crpo: A new approach for safe reinforcement learning with convergence guarantee.
>
> We have added the comparison results with IPO and CRPO to Appendix C.2.
>
> |Environment|IPO||CRPO||SAC-FDPI||
> |-|-|-|-|-|-|-|
> ||Cost|Return|Cost|Return|Cost|Return|
> |PointGoal|0.060|0.086|0.142|0.607|0.002|1.073|
> |PointPush|0.188|0.063|0.057|0.193|0.002|0.770|
> |PointCircle|0.052|0.816|0.013|0.918|0.004|0.797|
> |CarGoal|0.122|0.555|0.123|0.546|0.001|1.115|
> |CarPush|0.551|0.131|0.831|0.175|0.016|0.934|
> |CarCircle|0.045|0.624|0.025|0.677|0.003|0.619|
> |HalfCheetahVelocity|0.073|1.435|0.004|1.146|0.000|1.385|
> |HumanoidVelocity|0.001|0.162|0.002|0.635|0.000|0.810|
> |**Average**|0.136|0.484|0.149|0.612|0.003|0.938|
>
> ## Questions
>
> > My questions were resolved during the previous rebuttal phase of this paper, at an earlier conference. The only thing left: It would be nice if the additional results and discussions from the previous rebuttal can be added to the paper.
>
> Thank you for the suggestion. We have added all the results and discussions from the previous rebuttal have been added to the paper, including the sensitivity analysis for the KL divergence threshold $\delta$ in Appendix C.2.

---

> > ### Comment · Reviewer_W7n2 · 2025-11-27
> >
> > Thanks for the response and for updating the paper! I decide to retain my score.

---

### Official Review · Reviewer_opKu · 2025-11-01

**Soundness:** 3
**Presentation:** 3
**Contribution:** 2
**Rating:** 4
**Confidence:** 2

**Summary:**

This paper points out a safety paradox in safe RL: as a policy gets safer, it visits constraint-violating states less, making learning a feasibility function harder. The authors formalize this by showing the estimation error of the constraint decay function grows when violating samples become rare. To break this, the authors propose to run a primal policy that tries to be safe with a dual policy that seeks violations but is kept KL-close, and mix them with a truncated importance sampling.

**Strengths:**

1. The paper proposes a novel algorithm that uses a second KL-coupled policy to actively seek violating rollouts.
2. It is useful in the context of safe RL to formalize the safety paradox.

**Weaknesses:**

1. The bound that "error grows when violations are rarer" depends on the chosen CDF function, which is not motivated enough, making the paradox narrower than claimed.
2. FDPI creates unsafe data via the dual policy, and the paper does not quantify how many unsafe transitions it generates, or how to cap it in non-simulation settings.
3. The main difficulty, the state-distribution shift, is handled with a single-trajectory, truncated IS ratio plus a KL constraint. However, there is no bias/variance analysis on truncation depth, despite this is the core technical contribution.
4. The paper can be benefited from a broader scope of experiments. Currently all tasks are from Sarety-Gym with similar constraint types

**Questions:**

Please see weaknesses.

---

> ### Author Response · Authors · 2025-11-23
> **Authors' Response (Part 1)**
>
> We sincerely appreciate the reviewer's time and effort in reviewing our paper and providing such constructive feedback. Below, we address each of the concerns in detail.
>
> ## Weaknesses
>
> > The bound that "error grows when violations are rarer" depends on the chosen CDF function, which is not motivated enough, making the paradox narrower than claimed.
>
> We thank the reviewer for raising this important point. While our analysis is based on the CDF, it can be extended to more general settings. Here, we demonstrate the extension to the cost value function (CVF) widely used in CMDP, defined as $F^\pi(x)=\mathbb{E}\_{\tau\sim\pi}[\sum_{t=0}^\infty \gamma^t c(x_t)|x_0=x]$.
>
> The key insight is that the CVF can be decomposed into a discounted sum of CDF-like terms. Specifically, for any trajectory, we can segment it at constraint-violating states. The CVF is then equivalent to the discounted sum of the CDFs of these segmented sub-trajectories. Since our analysis shows that the estimation error bound increases for each of these constituent CDFs as the policy becomes safer, the error bound of the total CVF estimate must also increase.
>
> This confirms that the safety paradox not only holds for the specific CDF, but also applies to more general safe RL formulations. We have added this generalization to the remark at the end of Section 4 and Appendix D.2, where we also explained how our analysis can be extended from MC estimate to TD estimate.
>
> > FDPI creates unsafe data via the dual policy, and the paper does not quantify how many unsafe transitions it generates, or how to cap it in non-simulation settings.
>
> We thank the reviewer for raising this point. The dual threshold $d$ is the core mechanism that controls the volume of unsafe transitions. It acts as a setpoint for the data composition. When the average proportion of feasible states rises above $d$, the dual policy is activated to collect more infeasible states. Once the proportion falls back below $d$, the dual policy is deactivated. This creates a feedback loop that stabilizes the proportion of feasible states around $d$. The specific number of constraint violations is task-dependent but is inherently limited by this mechanism.
>
> Regarding non-simulation settings, there are two training and implementation modes in the broader safe RL community: (1) offline training and online deployment (OTOD), which first trains a policy in simulator and then deploys it in the real world, and (2) simultaneous online training and deployment (SOTD), which directly interacts with the real world to collect data for training. The OTOD mode only requires the final policy to be safe because intermediate policies will not be deployed in the real world. The SOTD model requires both the final policy and all intermediate policies to be safe. FDPI, along with the baselines we compare in the paper, belongs to the OTOD mode. We focus primarily on learning a safe policy at convergence, rather than guaranteeing safety during training. Therefore, FDPI does not involve unsafe exploration in the real world during training.
>
> We fully agree that for the SOTD mode, one must ensure safety throughout training. Achieving this goal would require integrating additional safe exploration techniques like those in [3,4]. These methods typically employ a model of the environment, either known or learned. Constraint violations are allowed in the model but not allowed in the environment. They alternate between learning a safe policy within the current model and refining the model with newly collected data. These approaches are complementary to our contribution, which focuses on solving the safe policy within a fixed model/environment.
>
> [3] Berkenkamp, F., et al. (2017). Safe model-based reinforcement learning with stability guarantees. Advances in neural information processing systems, 30.
>
> [4] Yu, D., et al. (2023). Safe model-based reinforcement learning with an uncertainty-aware reachability certificate. IEEE Transactions on Automation Science and Engineering, 21(3), 4129-4142.

---

> > ### Comment · Reviewer_opKu · 2025-11-27
> >
> > I thank the authors for the detailed response. I have reviewed their rebuttal, and it addresses most of my questions and concerns I raised in my reviews. I believe the paper can benefit from further improving the presentation of how results of CDF can be extended to constraint value functions in common CMDPs. I will update my ratings accordingly.

---

> ### Author Response · Authors · 2025-11-23
> **Authors' Response (Part 2)**
>
> > The main difficulty, the state-distribution shift, is handled with a single-trajectory, truncated IS ratio plus a KL constraint. However, there is no bias/variance analysis on truncation depth, despite this is the core technical contribution.
>
> We thank the reviewer for this comment. The single-trajectory sampling provides an unbiased estimate of the expected state-marginal ratio because each trajectory is an independent sample from the trajectory distribution. The IS ratio truncation also does not introduce bias because the truncation occurs on future states, which do not affect the probability of the current state's appearance in the trajectory.
>
> We would like to clarify that the IS truncation depth is not a tunable hyperparameter. The truncation is a direct consequence of the sampling process: for each state visited, its IS ratio is naturally computed from the initial state to its visitation. In other words, the truncation depth for each state equals the number of steps from the initial state to that state.
>
> Regarding the variance of the IS ratio, we have conducted an empirical analysis on PointGoal and AntVelocity. The IS ratios, which start from 1 at the beginning of training, stabilize to the following distributions well before the halfway point of the training process:
>
> |Environment|Mean|Variance|10th percentile|50th percentile|90th percentile|
> |-|-|-|-|-|-|
> |PointGoal|0.956|0.036|0.958|1.0|1.0|
> |AntVelocity|0.766|0.150|0.110|1.0|1.0|
>
> The results reveal two key insights. First, the fact that the median (50th percentile) IS ratio is 1.0 in both environments indicates that over half of the samples in the replay buffer are collected by the primal policy. This is expected, as the dual policy is not always active, and even when activated, half of the samples are still drawn from the primal policy. For these samples, the IS ratio is exactly 1.
>
> Second, the samples from the dual policy have IS ratios typically less than 1, as the probability of the same action is usually lower under the other policy. In PointGoal, the high 10th percentile (0.958) and low variance show that IS ratios are consistently close to 1, suggesting minimal IS correction is needed. In contrast, AntVelocity exhibits a lower 10th percentile (0.110) and higher variance, reflecting a greater proportion of samples from the dual policy. This aligns with the environment's dynamics, where a safer primal policy leads to more frequent activation of the dual policy.
>
> > The paper can be benefited from a broader scope of experiments. Currently all tasks are from Safety-Gym with similar constraint types.
>
> We thank the reviewer for this suggestion. We would like to clarify that our current evaluation already encompasses a diverse set of constraints within the Safety-Gymnasium benchmark, which extends beyond the original Safety-Gym to include velocity-constrained locomotion tasks. The constraint types we have tested are varied, including traversable hazards, impenetrable pillars, movable vases, dynamic moving obstacles, and velocity limits.
>
> To further strengthen our validation as per the reviewer's suggestion, we have conducted additional experiments on the SafetyHopper task from the Safety-MuJoCo benchmark [1]. This task introduces a fundamentally different constraint type: the robot must maintain a "healthy" state by keeping its height, angle, and other state elements within predefined safe ranges. This formulation is distinct from both the collision avoidance in our navigation tasks and the velocity constraints in our locomotion tasks, providing a complementary testbed for our method.
>
> |Algorithm|Cost|Return|
> |-|-|-|
> |CPO|212.527|-3202.929|
> |RCPO|71.817|1713.743|
> |FOCOPS|123.433|2637.076|
> |CUP|298.872|641.063|
> |PPO-Lag|122.778|2796.521|
> |SAC-Lag|98.240|974.359|
> |DSAC-T-Pen|11.550|2979.527|
> |SAC-FPI|5.350|3087.687|
> |SAC-FDPI|0.715|3138.275|
>
> Preliminary results (based on one seed) show that SAC-FDPI achieves the lowest cost and highest return. We are currently gathering results over multiple seeds and will include them in the paper once they are completed.
>
> [1] Gu, S., et al. (2024). Balance reward and safety optimization for safe reinforcement learning: A perspective of gradient manipulation. In Proceedings of the AAAI Conference on Artificial Intelligence (Vol. 38, No. 19, pp. 21099-21106).

---

> > ### Author Response · Authors · 2025-11-27
> >
> > The results for the SafetyHopper task are included in Table 2 in Appendix C.2. Please feel free to discuss if there is any further concern.

---

> ### Author Response · Authors · 2025-12-01
>
> We thank the reviewer for their positive feedback. We have improved the presentation of how results of CDF can be extended to CVF in Appendix D.2:
>
> Our analysis can also be extended to the CVF widely used in the CMDP, defined as
> $$
> F^\pi(x)=\mathbb{E}\_{\tau\sim\pi}\left[\sum\_{t=0}^\infty \gamma^t c(x_t)|x_0=x\right].
> $$
> The key insight is that the CVF can be decomposed into a discounted sum of CDF-like terms. Specifically, we can break down any infinite-horizon trajectory into segments that end immediately after a constraint violation. The total CVF is the discounted sum of costs along the entire trajectory. We can now group the costs by the segment in which they occur. The cost incurred in each segment is, by construction, a discounted sum that starts from an initial state and ends with a violation. Crucially, the value of the CVF in each segment is precisely a CDF. The total CVF can be expressed as:
> $$
> F^\pi(x) = \mathbb{E}[\text{discounted cost of Segment 1}] + \gamma^{T\_1}\mathbb{E}[\text{discounted cost of Segment 2}] + \cdots
> $$
> where $T\_i$ is the time step of the $i$th violation. Our main theoretical result establishes that the estimation error bound of a single CDF term increases as the policy becomes safer (i.e., as violations become rarer, making each segment longer). This directly implies that the error bound for the total CVF must also increase.

---

### Official Review · Reviewer_3DTX · 2025-11-06

**Soundness:** 3
**Presentation:** 3
**Contribution:** 3
**Rating:** 6
**Confidence:** 3

**Summary:**

The paper introduces a new theoretical problem in safe reinforcement learning (safe RL) called the safety paradox. This paradox arises because as an agent’s policy becomes safer—producing fewer constraint violations—it also generates fewer “unsafe” samples. However, those samples are crucial for accurately estimating the feasibility function, which measures whether future states will satisfy safety constraints. The result is a paradoxical loop: improving safety reduces the ability to learn safety accurately, which then harms the policy’s overall safety performance.

To address this issue, the authors propose a new algorithm called Feasible Dual Policy Iteration (FDPI). FDPI introduces an auxiliary “dual” policy that deliberately maximizes constraint violations while staying close to the original policy. This strategy increases the diversity and number of violating samples without requiring additional data collection. The algorithm then combines samples from both policies and corrects for the resulting distribution shift using importance sampling.

The authors conduct extensive experiments on the Safety-Gymnasium benchmark and show that FDPI achieves state-of-the-art results—producing the lowest constraint violation rates and competitive or best reward performance among all compared algorithms.

In essence, the paper both identifies a fundamental limitation in safe RL and provides a theoretically grounded, empirically validated solution to overcome it.

**Strengths:**

1.Novel Theoretical Insight — the “Safety Paradox”
The paper uncovers and rigorously explains a previously overlooked phenomenon in safe reinforcement learning: improving a policy’s safety reduces constraint-violating samples, which paradoxically increases the estimation error of the feasibility function and degrades future safety. This insight is both conceptually fresh and theoretically significant.

2.Principled and Effective Solution — Feasible Dual Policy Iteration (FDPI)
The proposed algorithm introduces a dual policy that deliberately generates controlled constraint violations to maintain balanced data for learning safety. With importance sampling and KL constraints ensuring stability, FDPI smartly breaks the paradox without altering the environment or reward structure.

3.Strong Empirical Validation
Extensive experiments on 14 Safety-Gymnasium environments demonstrate that FDPI achieves state-of-the-art safety performance (lowest violations) and competitive rewards. The empirical results clearly support the theoretical claims, showing reduced estimation error and consistent safety improvements across diverse tasks.

**Weaknesses:**

1.Too Many Theoretical Assumptions
The theoretical framework depends on several restrictive assumptions—such as smoothness of the distance-to-violation function, bounded state transitions, and weak covariance between trajectory segments. These simplify analysis but may not hold in high-dimensional or discontinuous environments, limiting the generality and real-world validity of the theory.

2.Limited Analysis of Hyperparameter Sensitivity
FDPI introduces new hyperparameters, including the dual activation threshold
d, KL divergence bound
δ, and feasibility tolerance
ϵ. However, the paper fixes these values without investigating their sensitivity or adaptive tuning. Since these parameters directly affect the balance between exploration and safety, a lack of analysis leaves uncertainty about robustness across different tasks.

3.Computational and Implementation Overhead
The algorithm maintains two policy networks and two feasibility estimators while applying importance-sampling corrections, which significantly increase computational and implementation complexity. The paper does not report runtime or convergence cost, raising concerns about scalability to larger or real-time safety-critical systems.

**Questions:**

1. Could you further explain the intuition behind the assumptions introduced in Section 4 (e.g., continuity of the distance-to-violation function, bounded state transitions, and weak covariance conditions)? It would be helpful to understand how these assumptions can still hold or approximately apply in practical scenarios, such as high-dimensional control or discontinuous dynamics.


2. While the motivation for Feasible Dual Policy Iteration (FDPI) is strong and well-argued, the paper does not appear to include a theoretical bound or convergence analysis for the proposed method. Could you clarify whether there exists any theoretical guarantee—such as stability, safety improvement, or sample-efficiency bounds—for FDPI, and how it connects to the theoretical framework established earlier?


3. In real-world applications where safety-critical systems cannot deliberately generate unsafe behaviors, how can the dual policy—designed to intentionally cause constraint violations—be implemented or approximated safely? Is the concept of a “dual policy” operationally feasible in physical or safety-limited environments, and what mechanisms could ensure it remains practical?

---

> ### Author Response · Authors · 2025-11-23
> **Authors' Response (Part 1)**
>
> We sincerely appreciate the reviewer's time and effort in reviewing our paper and providing such constructive feedback. Below, we address each of the concerns in detail.
>
> ## Weaknesses
>
> > **Too Many Theoretical Assumptions** The theoretical framework depends on several restrictive assumptions—such as smoothness of the distance-to-violation function, bounded state transitions, and weak covariance between trajectory segments. These simplify analysis but may not hold in high-dimensional or discontinuous environments, limiting the generality and real-world validity of the theory.
>
> We thank the reviewer for this critical comment. Our assumptions, while not universally applicable for every conceivable case, are valid for a vast majority of safety critical tasks. For high-dimensional or discontinuous environments, we do not require the entire state space to be continuous, only that the safety-related features used in the distance function, e.g., distance to obstacle and contact force, are continuous. This is achievable even in many high-dimensional hybrid systems.
>
> The smoothness of the distance function generally holds as long as the state contains enough information. For example, in a robotic manipulation task involving contact with fragile objects, one can define the distance function based on a gripper force sensor. As long as the force feedback is continuous, the distance to the constraint (e.g., a force threshold) will satisfy our assumption, despite the discrete nature of the contact event itself.
>
> Bounded state transitions is not an assumption we explicitly made, but is related to the smoothness of the distance function. The smoothness of the distance function implies that the transitions of the state features related to the distance function (not the entire state) are bounded. This is a mild and standard assumption in control and RL theory, and generally holds in physical systems with finite energy and control inputs.
>
> Regarding the weak covariance assumption, the core idea is that for a state very far from violation (with large $m(x)$), the initial step $N_1^\pi(x)$ is primarily influenced by local dynamics and the current state. In contrast, the subsequent long trajectory $\sum_{i=2}^{m(x)} N_i^\pi(x)$ is governed by future stochastic events. We assume that the correlation between this local initial step and the long, subsequent trajectory is weak relative to the variability of the initial step itself. This is reasonable in complex environments where the long-term path is highly unpredictable.
>
> > **Limited Analysis of Hyperparameter Sensitivity** FDPI introduces new hyperparameters, including the dual activation threshold $d$, KL divergence bound $\delta$, and feasibility tolerance $\epsilon$. However, the paper fixes these values without investigating their sensitivity or adaptive tuning. Since these parameters directly affect the balance between exploration and safety, a lack of analysis leaves uncertainty about robustness across different tasks.
>
> We thank the reviewer for this suggestion. We have now conducted sensitivity analyses on these hyperparameters. It is worth noting that these hyperparameters were not extensively tuned via grid search in the original paper.
>
> We evaluated the feasibility threshold $\epsilon\in \\{0.05,0.1,0.2\\}$ across 8 environments. The results indicate that a smaller $\epsilon$ leads to more conservative behavior, i.e., lower cost and lower return.
>
> |Environment|$\epsilon$=0.05||$\epsilon$=0.1||$\epsilon$=0.2||
> |-|-|-|-|-|-|-|
> ||Cost|Return|Cost|Return|Cost|Return|
> |PointGoal|0.001|0.998|0.002|1.073|0.017|1.080|
> |PointPush|0.000|0.822|0.002|0.770|0.001|0.816|
> |PointCircle|0.000|0.703|0.004|0.797|0.040|0.862|
> |CarGoal|0.000|1.087|0.001|1.115|0.000|1.085|
> |CarPush|0.001|0.862|0.016|0.934|0.008|0.888|
> |CarCircle|0.001|0.681|0.003|0.619|0.016|0.824|
> |HalfCheetahVelocity|0.000|1.386|0.000|1.385|0.000|1.418|
> |HumanoidVelocity|0.001|0.487|0.000|0.810|0.000|0.736|
> |**Average**|0.000|0.878|0.003|0.938|0.010|0.964|

---

> ### Author Response · Authors · 2025-11-23
> **Authors' Response (Part 2)**
>
> We evaluated the dual threshold $d\in \\{0.5,0.9,0.95,0.98\\}$. The results show that within a reasonable range (i.e., for $d\ge0.9$), a smaller $d$, which corresponds to more frequent activation of the dual policy, leads to lower costs without sacrificing return. However, an excessively small $d$ results in higher costs, possibly because of severe distributional shift.
>
> |Environment|$d$=0.50||$d$=0.90||$d$=0.95||$d$=0.98||
> |-|-|-|-|-|-|-|-|-|
> ||Cost|Return|Cost|Return|Cost|Return|Cost|Return|
> |PointGoal|0.001|0.988|0.000|1.091|0.002|1.073|0.008|1.076|
> |PointPush|0.003|0.747|0.000|0.746|0.002|0.770|0.005|0.798|
> |PointCircle|0.004|0.773|0.000|0.787|0.004|0.797|0.019|0.861|
> |CarGoal|0.003|1.107|0.000|1.142|0.001|1.115|0.000|1.110|
> |CarPush|0.150|0.943|0.005|1.028|0.016|0.934|0.021|0.789|
> |CarCircle|0.000|0.625|0.001|0.739|0.003|0.619|0.001|0.703|
> |HalfCheetahVelocity|0.000|1.374|0.000|1.382|0.000|1.385|0.000|1.374|
> |HumanoidVelocity|0.000|0.825|0.000|0.682|0.000|0.810|0.000|0.763|
> |**Average**|0.020|0.923|0.001|0.949|0.003|0.938|0.007|0.934|
>
> We evaluated the KL threshold $\delta\in \\{2,5,10\\}$. The results show that the overall performance is stable within a reasonable range ($\delta\le5$), with a smaller $\delta$ slightly decreases both cost and return. However, an excessively large $\delta$ significantly increases cost due to excessive policy divergence.
>
> |Environment|$\delta$=2||$\delta$=5||$\delta$=10||
> |-|-|-|-|-|-|-|
> ||Cost|Return|Cost|Return|Cost|Return|
> |PointGoal|0.002|1.000|0.002|1.073|0.015|1.036|
> |PointPush|0.000|0.335|0.002|0.770|0.061|0.647|
> |PointCircle|0.005|0.775|0.004|0.797|0.012|0.861|
> |CarGoal|0.000|1.108|0.001|1.115|0.002|1.144|
> |CarPush|0.010|0.857|0.016|0.934|0.387|0.021|
> |CarCircle|0.002|0.766|0.003|0.619|0.005|0.824|
> |HalfCheetahVelocity|0.000|1.433|0.000|1.385|0.000|1.425|
> |HumanoidVelocity|0.000|0.843|0.000|0.810|0.000|0.709|
> |**Average**|0.002|0.890|0.003|0.938|0.060|0.833|
>
> These results have been added to Appendix C.2.
>
> > **Computational and Implementation Overhead** The algorithm maintains two policy networks and two feasibility estimators while applying importance-sampling corrections, which significantly increase computational and implementation complexity. The paper does not report runtime or convergence cost, raising concerns about scalability to larger or real-time safety-critical systems.
>
> We thank the reviewer for this comment. We explicitly measured the training cost of SAC-FPI (single policy) and SAC-FDPI on 6 representative environments (4 navigation tasks and 2 locomotion tasks).
>
> |Environment|SAC-FPI||SAC-FDPI||
> |-|-|-|-|-|
> ||Time (s)|Param # (K)|Time (s)|Param # (K)|
> |PointGoal|8174|739|8469|1150|
> |PointPush|8148|739|8341|1150|
> |PointButton|9826|739|9955|1150|
> |PointCircle|8827|739|9501|1150|
> |HalfCheetahVelocity|5066|651|5614|1013|
> |HumanoidVelocity|5760|1506|6189|2345|
> |**Average**|7634|852|8012|1327|
>
> The average training time of SAC-FDPI increases by about 5% compared with that of SAC-FPI, and the number of parameters increases by about 56%. This indicates that although FDPI has an additional policy and feasibility network, the computational time does not increase much. This is because (1) the dual policy does not increase the total number of environmental interactions, i.e., samples collected by the dual policy are counted into the total samples, and (2) the loss function of the dual policy only involves the dual feasibility network, thus its backpropagation is more computationally efficient than the primal policy, which includes both the Q network and feasibility network.
>
> The reason that the number of parameters increases by a large proportion is that we use separate networks for the Q value, feasibility value, and policy. In high-dimensional tasks with multi-modal observations, it is common practice to use a shared backbone for Q value, feasibility, and policy. In this case, the dual policy and dual feasibility network are just two additional heads, which account for a small proportion of the total parameters.

---

> ### Author Response · Authors · 2025-11-23
> **Authors' Response (Part 3)**
>
> ## Questions
>
> > Could you further explain the intuition behind the assumptions introduced in Section 4 (e.g., continuity of the distance-to-violation function, bounded state transitions, and weak covariance conditions)? It would be helpful to understand how these assumptions can still hold or approximately apply in practical scenarios, such as high-dimensional control or discontinuous dynamics.
>
> See our response to Weakness 1.
>
> > While the motivation for Feasible Dual Policy Iteration (FDPI) is strong and well-argued, the paper does not appear to include a theoretical bound or convergence analysis for the proposed method. Could you clarify whether there exists any theoretical guarantee—such as stability, safety improvement, or sample-efficiency bounds—for FDPI, and how it connects to the theoretical framework established earlier?
>
> We thank the reviewer for this insightful question. FDPI is a direct and practical implementation derived from our core theoretical analysis. While providing a full end-to-end convergence proof for the complete FDPI loop is a challenging and open problem, the method is built upon a solid theoretical foundation that guarantees its core mechanism.
>
> Specifically, Theorems 1 and 2 establish that the CDF estimation error is bounded by the variance of the steps to violation, and that this variance increases with policy safety. By using a dual policy to actively collect more violating states, FDPI directly counteracts the increasing sparsity proven in Theorem 2, thereby reducing the CDF estimation error bound implied by Theorem 1. In this way, FDPI's theoretical guarantee lies in its principled mitigation of the key bottleneck we identified. A formal sample complexity or convergence analysis is a highly valuable direction for future work.
>
> > In real-world applications where safety-critical systems cannot deliberately generate unsafe behaviors, how can the dual policy—designed to intentionally cause constraint violations—be implemented or approximated safely? Is the concept of a "dual policy" operationally feasible in physical or safety-limited environments, and what mechanisms could ensure it remains practical?
>
> We thank the reviewer for raising this important concern. In the broader safe RL community, there are two training and implementation modes: (1) offline training and online deployment (OTOD), which first trains a policy in simulator and then deploys it in the real world, and (2) simultaneous online training and deployment (SOTD), which directly interacts with the real world to collect data for training. The OTOD mode only requires the final policy to be safe because intermediate policies will not be deployed in the real world. The SOTD model requires both the final policy and all intermediate policies to be safe.
>
> FDPI, along with the baselines we compare in the paper, belongs to the OTOD mode. We focus primarily on learning a safe policy at convergence, rather than guaranteeing safety during training. Therefore, FDPI does not involve unsafe exploration in the real world during training.
>
> We fully agree that for the SOTD mode, one must ensure safety throughout training. Achieving this goal would require integrating additional safe exploration techniques like those in [3,4]. These methods typically employ a model of the environment, either known or learned. Constraint violations are allowed in the model but not allowed in the environment. They alternate between learning a safe policy within the current model and refining the model with newly collected data. These approaches are complementary to our contribution, which focuses on solving the safe policy within a fixed model/environment.
>
> [3] Berkenkamp, F., et al. (2017). Safe model-based reinforcement learning with stability guarantees. Advances in neural information processing systems, 30.
>
> [4] Yu, D., et al. (2023). Safe model-based reinforcement learning with an uncertainty-aware reachability certificate. IEEE Transactions on Automation Science and Engineering, 21(3), 4129-4142.

---

### Official Review · Reviewer_2igd · 2025-11-06

**Soundness:** 4
**Presentation:** 4
**Contribution:** 4
**Rating:** 8
**Confidence:** 2

**Summary:**

The paper identifies and formalizes a **safety paradox**: as a policy becomes safer, **violation samples become sparse**, which makes the **feasibility function** (e.g., a CDF) harder to estimate and actually increases its error bound—degrading feasible-set identification and, paradoxically, safety. To break this loop, the authors propose **FDPI (Feasible Dual Policy Iteration)**: alongside the primal policy, a **dual policy** deliberately generates violation samples; data from both policies are **importance-sampled (IS)** to correct distribution shift, and a **(symmetric) KL constraint** stabilizes the IS ratios. On 14 Safety-Gymnasium tasks, FDPI achieves **the lowest violation cost** while maintaining **best or near-best return**.

**Strengths:**

* **Clear, broadly relevant problem framing**: Elevates the “sparse violations → higher feasibility error → less safety” feedback loop into a unifying paradox for feasibility-function–based safe RL.
* **Coherent method design**: The **dual policy** raises violation coverage without increasing total interactions; **truncated-trajectory IS** approximates the state-marginal ratio, and a **KL constraint** tames ratio underflow—practically implementable.
* **Compatible with common policy-improvement frameworks** (e.g., FPI/SAC) with concrete loss forms and Lagrangian updates; reproducibility looks feasible from the described components.
* **Empirical scope and metrics are solid**: 14 environments with cost–return trade-off plots showing **lowest cost** and **competitive returns**, plus targeted ablations that match the stated research questions.

**Weaknesses:**

* **Theory–practice gap**: Error bounds are developed under MC/CDF estimation, while training uses **TD learning with function approximation**; the bridge between them is under-substantiated.
* **Assumption generality**: Key results rely on continuity/neighboring-state bounds and other conditions whose validity in contact-rich or discontinuous dynamics is not fully justified.
* **IS approximation and robustness**: The state-marginal ratio is approximated along a single trajectory and truncated at first appearance, which can bias estimates; there’s no systematic **sensitivity study** for the **KL threshold, truncation length, or dual sampling fraction**.

**Questions:**

1. Can you provide an **error-propagation analysis under TD learning**, or at least a joint training trace that links **violation coverage → feasibility-estimation error → final cost**, to empirically validate the paradox-breaking story?
2. Please include **sensitivity analyses** for the **KL threshold**, **dual activation/sampling ratio**, and **IS truncation length**; also report the **distribution of IS ratios** (mean/variance/overflow/underflow rates) over training.

---

> ### Author Response · Authors · 2025-11-23
> **Authors' Response (Part 1)**
>
> We sincerely appreciate the reviewer's time and effort in reviewing our paper and providing such constructive feedback. Below, we address each of the concerns in detail.
>
> ## Weaknesses
>
> > **Theory–practice gap**: Error bounds are developed under MC/CDF estimation, while training uses TD learning with function approximation; the bridge between them is under-substantiated.
>
> We thank the reviewer for this observation. The remark at the end of Section 4 notes that our analysis, while based on MC estimate, extends to TD estimate. The core reason is that the increased variance identified in Theorem 2 propagates to the TD target. Specifically, Section 4.2 proves that the variance of steps to violation, $N(\tau)$, increases with safer policies. Consider an initial state $x$, its subsequent state $x'$, and the sub-trajectory $\tau'$ starting from $x'$. Since $N(\tau')=N(\tau)-1$, the variance of $N(\tau')$ also increases with policy safety. This implies a higher variance in the true feasibility value $F^\pi(x')$. Consequently, the TD target, which is computed by $F^\pi(x')$, inherits this increased variance, leading to a larger estimation error in $\hat{F}^\pi(x)$. We acknowledge that this connection could have been made more explicit. We have revised the remark at the end of Section 4 and added a detailed explanation to the Appendix D.2 to clarify this point.
>
> The reviewer is correct that function approximation introduces an additional source of error. Our analysis focuses on the data-centric cause of estimation error, i.e., the increasing sparsity of violating samples. Function approximation error is a separate, model-centric challenge. For any approximation function, the data-centric challenge of learning from increasingly sparse signal remains. We defer the analysis of approximation error to future work.
>
> > **Assumption generality**: Key results rely on continuity/neighboring-state bounds and other conditions whose validity in contact-rich or discontinuous dynamics is not fully justified.
>
> We thank the reviewer for raising this important point. It is true that Assumption 3 (continuity of distance) requires the distance function to change smoothly with the dynamics. While this may not hold for every conceivable case, it is valid for a vast majority of practical applications. Crucially, this assumption often holds even in contact-rich tasks through careful problem formulation. For instance, in a robotic manipulation task involving contact with fragile objects, one can define the distance function based on a continuous sensor reading, such as the gripper force. As long as the force feedback is continuous, the distance to the constraint (e.g., a force threshold) will satisfy our assumption, despite the discrete nature of the contact event itself. Therefore, the assumption remains widely applicable when the safety constraint is properly defined using available continuous signals.
>
> > **IS approximation and robustness**: The state-marginal ratio is approximated along a single trajectory and truncated at first appearance, which can bias estimates; there’s no systematic sensitivity study for the KL threshold, truncation length, or dual sampling fraction.
>
> We thank the reviewer for this question. The single-trajectory sampling provides an unbiased estimate of the expected state-marginal ratio because each trajectory is an independent sample from the trajectory distribution. The IS ratio truncation also does not introduce bias because the truncation occurs on future states, which do not affect the probability of the current state's appearance in the trajectory.
>
> We would like to clarify that the IS truncation length is not a tunable hyperparameter. The truncation is a direct consequence of the sampling process: for each state visited, its IS ratio is naturally computed from the initial state to its visitation. In other words, the truncation length for each state equals the number of steps from the initial state to that state.
>
> We have added a sensitivity study for the KL threshold $\delta$. We evaluated $\delta\in \\{2,5,10\\}$. The results show that the overall performance is stable within a reasonable range ($\delta\le5$), with a smaller $\delta$ slightly decreases both cost and return. However, an excessively large $\delta$ significantly increases cost due to excessive policy divergence.
>
> |Environment|$\delta$=2||$\delta$=5||$\delta$=10||
> |-|-|-|-|-|-|-|
> ||Cost|Return|Cost|Return|Cost|Return|
> |PointGoal|0.002|1.000|0.002|1.073|0.015|1.036|
> |PointPush|0.000|0.335|0.002|0.770|0.061|0.647|
> |PointCircle|0.005|0.775|0.004|0.797|0.012|0.861|
> |CarGoal|0.000|1.108|0.001|1.115|0.002|1.144|
> |CarPush|0.010|0.857|0.016|0.934|0.387|0.021|
> |CarCircle|0.002|0.766|0.003|0.619|0.005|0.824|
> |HalfCheetahVelocity|0.000|1.433|0.000|1.385|0.000|1.425|
> |HumanoidVelocity|0.000|0.843|0.000|0.810|0.000|0.709|
> |**Average**|0.002|0.890|0.003|0.938|0.060|0.833|

---

> ### Author Response · Authors · 2025-11-23
> **Authors' Response (Part 2)**
>
> We also added a sensitivity study for the dual threshold $d$, which controls the dual sampling fraction. We evaluated $d\in \\{0.5,0.9,0.95,0.98\\}$. The results show that within a reasonable range (i.e., for $d\ge0.9$), a smaller $d$, which corresponds to more frequent activation of the dual policy, leads to lower costs without sacrificing return. However, an excessively small $d$ results in higher costs, possibly because of severe distributional shift.
>
> |Environment|$d$=0.50||$d$=0.90||$d$=0.95||$d$=0.98||
> |-|-|-|-|-|-|-|-|-|
> ||Cost|Return|Cost|Return|Cost|Return|Cost|Return|
> |PointGoal|0.001|0.988|0.000|1.091|0.002|1.073|0.008|1.076|
> |PointPush|0.003|0.747|0.000|0.746|0.002|0.770|0.005|0.798|
> |PointCircle|0.004|0.773|0.000|0.787|0.004|0.797|0.019|0.861|
> |CarGoal|0.003|1.107|0.000|1.142|0.001|1.115|0.000|1.110|
> |CarPush|0.150|0.943|0.005|1.028|0.016|0.934|0.021|0.789|
> |CarCircle|0.000|0.625|0.001|0.739|0.003|0.619|0.001|0.703|
> |HalfCheetahVelocity|0.000|1.374|0.000|1.382|0.000|1.385|0.000|1.374|
> |HumanoidVelocity|0.000|0.825|0.000|0.682|0.000|0.810|0.000|0.763|
> |**Average**|0.020|0.923|0.001|0.949|0.003|0.938|0.007|0.934|
>
> These results have been added to Appendix C.2.
>
> ## Questions
>
> > Can you provide an error-propagation analysis under TD learning, or at least a joint training trace that links violation coverage → feasibility-estimation error → final cost, to empirically validate the paradox-breaking story?
>
> We thank the reviewer for the insightful question. The extension from MC to TD is provided in our response to Weakness 1.
>
> An empirical validation of the paradox-breaking chain can be seen by combining Fig. 3, 4, and 1 in our paper. Specifically, Fig. 3 shows that SAC-FDPI maintains about 10× more violating samples in its replay buffer compared to SAC-FPI in most environments. This richer violation data directly leads to the outcome shown in Fig. 4: SAC-FDPI achieves significantly lower estimation errors than SAC-FPI in the same tasks. This inverse relationship between violating sample count and estimation error supports our theoretical analysis: richer violation data leads to better feasibility estimation. The consequence of this accurate estimation is shown in Fig. 1: SAC-FDPI achieves a substantially lower final cost than SAC-FPI while maintaining a competitive return.
>
> > Please include sensitivity analyses for the KL threshold, dual activation/sampling ratio, and IS truncation length; also report the distribution of IS ratios (mean/variance/overflow/underflow rates) over training.
>
> We thank the reviewer for this suggestion. We have now included sensitivity analyses for the KL threshold and the dual threshold in our response to Weakness 3, along with an explanation for the IS truncation length.
>
> Regarding the distribution of the IS ratio, we have conducted an empirical analysis on PointGoal and AntVelocity. The IS ratios, which start from 1 at the beginning of training, stabilize to the following distributions well before the halfway point of the training process:
>
> |Environment|Mean|Variance|10th percentile|50th percentile|90th percentile|
> |-|-|-|-|-|-|
> |PointGoal|0.956|0.036|0.958|1.0|1.0|
> |AntVelocity|0.766|0.150|0.110|1.0|1.0|
>
> The results reveal two key insights. First, the fact that the median (50th percentile) IS ratio is 1.0 in both environments indicates that over half of the samples in the replay buffer are collected by the primal policy. This is expected, as the dual policy is not always active, and even when activated, half of the samples are still drawn from the primal policy. For these samples, the IS ratio is exactly 1.
>
> Second, the samples from the dual policy have IS ratios typically less than 1, as the probability of the same action is usually lower under the other policy. In PointGoal, the high 10th percentile (0.958) and low variance show that IS ratios are consistently close to 1, suggesting minimal IS correction is needed. In contrast, AntVelocity exhibits a lower 10th percentile (0.110) and higher variance, reflecting a greater proportion of samples from the dual policy. This aligns with the environment's dynamics, where a safer primal policy leads to more frequent activation of the dual policy.

---

### Official Review · Reviewer_HJpx · 2025-11-07

**Soundness:** 4
**Presentation:** 3
**Contribution:** 3
**Rating:** 6
**Confidence:** 4

**Summary:**

This paper identifies the “safety paradox” in safe reinforcement learning, where making a policy safer reduces constraint-violating samples and thus worsens safety estimation. It proposes Feasible Dual Policy Iteration (FDPI), which adds a dual policy that deliberately induces controlled violations to improve feasibility function accuracy using importance sampling and KL constraints. Empirical experiments show FDPI achieves strong safety and returns, breaking the paradox

**Strengths:**

* **Problem Significance:** The paper addresses the well-known and critical problem of learning from sparse cost signals in safe R. Accurately modeling safety boundaries, especially when violations are rare , is a key obstacle to deploying RL in the real world
* **Writing:** This paper is in general well written, barring some minor issues, see weaknesses and questions for a detailed list.
Intuitive Solution: The proposed mechanism of using a dedicated "dual policy" to actively generate the very data that is missing (i.e., constraint violations) is an intuitive and direct approach to solving the sparse data problem.
* **Experimental Results:** The experimental results presented in Figure 1 and 2 show that FDPI (and its ablative variant, SAC-FPI) achieve a superior cost/return trade-off when compared against the chosen set of baselines.

**Weaknesses:**

1. **Missing Context of the "Safety Paradox.”** At the crux, this paradox is due to the sparsity of the safety constraint, which becomes more sparse as the policy becomes more sparse. The authors address this for one specific policy iteration algorithm, FPI, by using a dual policy to sample unsafe data. There have been existing works that tackle this problem from a different angle. For example “Feasibility Consistent Representation Learning for Safe Reinforcement Learning”[1] proposes using representation learning to extract the safety-related information from the raw state. Perhaps the authors could include a discussion or comparisons to such methods.
2. **Baselines:** The current suite of baselines are limited and rather old, with the newest being from 2023\. Perhaps some newer baselines could be added, such as ESPO [2] and PCRPO [3], and the FCSRL[1] method mentioned above?
3. **Ablations:** There are no ablations of the key design hyperparameters such as $\\epsilon$ and $d$.
4. **Over usage of $d$ (Minor)** $d$ is used several times in the paper for different notations. For example, d is defined as a continuous function that measures the distance to a constraint violation in Assumption 2\. But then in lines 259-260, it is defined again as a dual threshold hyperparameter.
5. **Intuitive Justification of Assumption 4 (Minor)** The intuitive justification for Assumption 4 is not particularly clear.

[1] Cen, Zhepeng, et al. "Feasibility consistent representation learning for safe reinforcement learning." Proceedings of the 41st International Conference on Machine Learning. 2024.

[2] Gu, Shangding, et al. "Enhancing efficiency of safe reinforcement learning via sample manipulation." Advances in Neural Information Processing Systems 37 (2024): 17247-17285.

[3] Gu, Shangding, et al. "Balance reward and safety optimization for safe reinforcement learning: A perspective of gradient manipulation." Proceedings of the AAAI Conference on Artificial Intelligence. Vol. 38. No. 19. 2024.

**Questions:**

1. What is the impact of the sampling fraction from the dual policy? In the paper, it seems to be fixed at 0.5. In particular, if compute allows, an interesting ablation would be how altering both the dual threshold and sampling fraction would impact the final policy.
2. In Eq 5, the optimization problem is formulated with a constraint that the expectation of the CDF is less than 0, ie $\\mathbb{E}\_{x\\sim d\_{init}}\\left\[F^{\\pi}(x)\\right\]\\leq 0$ However in equation 4, the CDF is defined as the expectation of $\\gamma^{N(\\tau)}$ where $\\gamma \\in (0,1)$, so how is constraint possible?
3. Could the authors clarify the precise relationship with FPI? It appears the core primal policy update (Eq. 11\) is adopted from FPI, and the primary novelty is the dual-policy data-augmentation mechanism designed to fix FPI's data-starvation failure mode (the paradox). Is this an accurate characterization of the contribution?
4. Are the proposed data centric solutions and existing representation centric solutions mutually exclusive? Could FDPI's dual-policy sampler be combined with a safety-aware embedding (like that from FCSRL) to potentially achieve even greater stability and performance?

---

> ### Author Response · Authors · 2025-11-23
> **Authors' Response (Part 1)**
>
> We sincerely appreciate the reviewer's time and effort in reviewing our paper and providing such constructive feedback. Below, we address each of the concerns in detail.
>
> ## Weaknesses
>
> > **Missing Context of the "Safety Paradox"** At the crux, this paradox is due to the sparsity of the safety constraint, which becomes more sparse as the policy becomes more sparse. The authors address this for one specific policy iteration algorithm, FPI, by using a dual policy to sample unsafe data. There have been existing works that tackle this problem from a different angle. For example "Feasibility Consistent Representation Learning for Safe Reinforcement Learning" [1] proposes using representation learning to extract the safety-related information from the raw state. Perhaps the authors could include a discussion or comparisons to such methods.
> >
> > [1] Cen, Zhepeng, et al. "Feasibility consistent representation learning for safe reinforcement learning." Proceedings of the 41st International Conference on Machine Learning. 2024.
>
> We appreciate the suggestion to discuss FCSRL [1], which also addresses the challenge of sparse violation signals in feasibility function estimation. FCSRL focuses on representation learning, which tries to better exploit safety-related information from given data. In contrast, our method addresses the safety paradox, where violation signals become increasingly sparse as the policy becomes safer. Our solution is to actively collect more violating data using a dual policy, targeting the root cause of the sparsity. Therefore, FCSRL and our method are orthogonal: FCSRL improves data utilization, while our method improves data collection. We have added a discussion of this distinction in Section 2 and Appendix D.1.
>
> > **Baselines** The current suite of baselines are limited and rather old, with the newest being from 2023. Perhaps some newer baselines could be added, such as ESPO [2] and PCRPO [3], and the FCSRL[1] method mentioned above?
> >
> > [2] Gu, Shangding, et al. "Enhancing efficiency of safe reinforcement learning via sample manipulation." Advances in Neural Information Processing Systems 37 (2024): 17247-17285.
> >
> > [3] Gu, Shangding, et al. "Balance reward and safety optimization for safe reinforcement learning: A perspective of gradient manipulation." Proceedings of the AAAI Conference on Artificial Intelligence. Vol. 38. No. 19. 2024.
>
> We thank the reviewer for this suggestion. We have added FCSRL [1] for comparison on all environments. Initial results (based on one seed) show that the averaged cost and return of SAC-FDPI are both better than FCSRL. We are currently gathering results over multiple seeds and will include them in the paper once they are completed.
>
> |Environment|FCSRL||SAC-FDPI||
> |-|-|-|-|-|
> ||Cost|Return|Cost|Return|
> |PointGoal|0.022|0.625|0.002|1.073|
> |PointPush|0.000|0.573|0.002|0.770|
> |PointButton|0.000|0.008|0.006|0.102|
> |PointCircle|0.029|0.266|0.004|0.797|
> |CarGoal|0.000|1.018|0.001|1.115|
> |CarPush|0.019|0.719|0.016|0.934|
> |CarButton|0.008|-0.072|0.013|0.066|
> |CarCircle|0.036|0.012|0.003|0.619|
> |AntVelocity|0.001|0.518|0.000|0.469|
> |HumanoidVelocity|0.013|0.159|0.000|0.810|
> |HalfCheetahVelocity|0.001|1.373|0.000|1.385|
> |HopperVelocity|0.001|0.760|0.000|0.801|
> |Walker2dVelocity|0.001|0.551|0.000|0.579|
> |SwimmerVelocity|0.011|0.315|0.000|0.355|
> |**Average**|0.010|0.488|0.003|0.705|
>
> We also actively sought to include ESPO [2] and PCRPO [3]. However, despite our efforts, we could not find official or third-party implementations for these methods. We will gladly include them in future comparisons once the code is released.

---

> ### Author Response · Authors · 2025-11-23
> **Authors' Response (Part 2)**
>
> > **Ablations** There are no ablations of the key design hyperparameters such as $\epsilon$ and $d$.
>
> We thank the reviewer for this suggestion. We have now conducted ablation studies on these hyperparameters. It is worth noting that these hyperparameters were not extensively tuned via grid search in the original paper.
>
> We evaluated $\epsilon\in \\{0.05,0.1,0.2\\}$ across 8 environments. The results indicate that a smaller $\epsilon$ leads to more conservative behavior, i.e., lower cost and lower return.
>
> |Environment|$\epsilon$=0.05||$\epsilon$=0.1||$\epsilon$=0.2||
> |-|-|-|-|-|-|-|
> ||Cost|Return|Cost|Return|Cost|Return|
> |PointGoal|0.001|0.998|0.002|1.073|0.017|1.080|
> |PointPush|0.000|0.822|0.002|0.770|0.001|0.816|
> |PointCircle|0.000|0.703|0.004|0.797|0.040|0.862|
> |CarGoal|0.000|1.087|0.001|1.115|0.000|1.085|
> |CarPush|0.001|0.862|0.016|0.934|0.008|0.888|
> |CarCircle|0.001|0.681|0.003|0.619|0.016|0.824|
> |HalfCheetahVelocity|0.000|1.386|0.000|1.385|0.000|1.418|
> |HumanoidVelocity|0.001|0.487|0.000|0.810|0.000|0.736|
> |**Average**|0.000|0.878|0.003|0.938|0.010|0.964|
>
> We evaluated $d\in \\{0.5,0.9,0.95,0.98\\}$. The results show that within a reasonable range (i.e., for $d\ge0.9$), a smaller $d$, which corresponds to more frequent activation of the dual policy, leads to lower costs without sacrificing return. However, an excessively small $d$ results in higher costs, possibly because of severe distributional shift.
>
> |Environment|$d$=0.50||$d$=0.90||$d$=0.95||$d$=0.98||
> |-|-|-|-|-|-|-|-|-|
> ||Cost|Return|Cost|Return|Cost|Return|Cost|Return|
> |PointGoal|0.001|0.988|0.000|1.091|0.002|1.073|0.008|1.076|
> |PointPush|0.003|0.747|0.000|0.746|0.002|0.770|0.005|0.798|
> |PointCircle|0.004|0.773|0.000|0.787|0.004|0.797|0.019|0.861|
> |CarGoal|0.003|1.107|0.000|1.142|0.001|1.115|0.000|1.110|
> |CarPush|0.150|0.943|0.005|1.028|0.016|0.934|0.021|0.789|
> |CarCircle|0.000|0.625|0.001|0.739|0.003|0.619|0.001|0.703|
> |HalfCheetahVelocity|0.000|1.374|0.000|1.382|0.000|1.385|0.000|1.374|
> |HumanoidVelocity|0.000|0.825|0.000|0.682|0.000|0.810|0.000|0.763|
> |**Average**|0.020|0.923|0.001|0.949|0.003|0.938|0.007|0.934|
>
> These results have been added to Appendix C.2.
>
> > **Over usage of $d$ (Minor)** $d$ is used several times in the paper for different notations. For example, $d$ is defined as a continuous function that measures the distance to a constraint violation in Assumption 2. But then in lines 259-260, it is defined again as a dual threshold hyperparameter.
>
> We thank the reviewer for pointing out this notational overlap. To resolve the ambiguity, we have updated the manuscript as follows: (1) the distance function is now denoted as $D$, and (2) the marginal state distribution of policy $\pi$ is now denoted as $p^\pi$.
>
> > **Intuitive Justification of Assumption 4 (Minor)** The intuitive justification for Assumption 4 is not particularly clear.
>
> We thank the reviewer for this feedback. We agree that the intuition for Assumption 4 can be elaborated. The core idea is that for a state very far from violation (with large $m(x)$), the initial step $N_1^\pi(x)$ is primarily influenced by local dynamics and the current state. In contrast, the subsequent long trajectory $\sum_{i=2}^{m(x)} N_i^\pi(x)$ is governed by future stochastic events. We assume that the correlation between this local initial step and the long, subsequent trajectory is weak relative to the variability of the initial step itself. This is reasonable in complex environments where the long-term path is highly unpredictable. We have revised the justification for this assumption in the manuscript.

---

> ### Author Response · Authors · 2025-11-23
> **Authors' Response (Part 3)**
>
> ## Questions
>
> > What is the impact of the sampling fraction from the dual policy? In the paper, it seems to be fixed at 0.5. In particular, if compute allows, an interesting ablation would be how altering both the dual threshold and sampling fraction would impact the final policy.
>
> We thank the reviewer for this insightful question. Our design choice to fix the sampling fraction at 0.5 was intentional, as both hyperparameters fundamentally control the same thing: the proportion of data from the dual policy. We found the dual threshold to be a more principled and interpretable controller for this proportion. It has a direct relationship with the average proportion of feasible states. When the proportion rises above $d$, the dual policy is activated to collect more infeasible states. Once the proportion falls back below $d$, the dual policy is deactivated. This creates a self-regulating data collection loop that is responsive to the policy's current safety performance.
>
> The ablation study of $d$ is provided in our response to Weakness 3.
>
> >In Eq 5, the optimization problem is formulated with a constraint that the expectation of the CDF is less than 0, i.e., $\mathbb{E}\_{x \sim d_{init}}[F^\pi(x)] \le 0$. However in equation 4, the CDF is defined as the expectation of $\gamma^{N(\tau)}$ where $\gamma\in(0,1)$, so how is constraint possible?
>
> We thank the reviewer for this observation. The constraint $\mathbb{E}\_{x\sim d_\text{init}}[F^\pi(x)] \le 0$ is indeed feasible and aligns with the mathematical formulation of our goal: strict zero violation. The key is that the steps to violation $N(\tau)$ can be $+\infty$ for a trajectory that never violates a constraint. The notation $N(\tau)\in\mathbb{N}$ in the original paper is imprecise. We have changed it to $N(\tau)\in\mathbb{N}\cup\\{+\infty\\}$.
>
> > Could the authors clarify the precise relationship with FPI? It appears the core primal policy update (Eq. 11) is adopted from FPI, and the primary novelty is the dual-policy data-augmentation mechanism designed to fix FPI's data-starvation failure mode (the paradox). Is this an accurate characterization of the contribution?
>
> We thank the reviewer for this question. The reviewer's characterization captures an important aspect of our work: our primal policy update builds upon FPI's principles, and a key innovation is the dual-policy mechanism that addresses data starvation. However, FDPI is more than "FPI plus a dual-policy data-augmentation trick."
>
> Theoretically, we identify and formalize the safety paradox, a core challenge in safe RL where improved policy safety provably increases the estimation error bound of the feasibility function due to vanishing constraint-violating data. This reveals a concrete limitation of existing safe RL methods, including FPI, and motivates our design.
>
> Algorithmically, we introduce a dual policy and dual feasibility function that explicitly maximize constraint violations to maintain a controlled amount of unsafe data. To safely use dual-policy data, we use IS to reweight the loss functions for both the primal and the dual policy. We further impose KL constraints to keep IS ratios stable while allowing sufficient dual-policy divergence.
>
> > Are the proposed data centric solutions and existing representation centric solutions mutually exclusive? Could FDPI's dual-policy sampler be combined with a safety-aware embedding (like that from FCSRL) to potentially achieve even greater stability and performance?
>
> These two kinds of methods are orthogonal and complementary, not mutually exclusive. As we mentioned in our response to Weakness 1, representation centric methods focus on extacting better features from given data, while our data centric method focuses on actively collecting more critical samples. Combining FDPI's dual policy mechanism with a safety-aware encoder from FCSRL is a highly promising direction. The improved representation could allow for even more efficient policy optimization on the enriched data provided by our method.

---

> > ### Comment · Reviewer_HJpx · 2025-11-23
> >
> > Thank you for the detailed rebuttal. I'm convinced by the discussion, and have raised my score.

---

> ### Author Response · Authors · 2025-11-27
>
> Thank you for your positive feedback and for raising the score. We are delighted to hear that you found our rebuttal convincing. The results for FCSRL are included in Table 2 in Appendix C.2.

---

### Official Review · Reviewer_EGeQ · 2025-11-11

**Soundness:** 3
**Presentation:** 3
**Contribution:** 3
**Rating:** 4
**Confidence:** 3

**Summary:**

This paper studies safe RL. It first introduces a problem termed as safety paradox, which describes a phenomenon that as a safe RL agent's policy improves, it encounters constraint-violating states less frequently. The consequence of this paradox is that an inaccurate feasibility function estimate leads to an incorrect identification of the feasible region. This biases the policy update and undermines policy safety. To address it, the paper proposes an algorithm called Feasible Dual Policy Iteration. The algorithm maintains and trains two policies in parallel. The dual policy is trained to maximize constraint violations by maximizing the dual feasibility function. This mechanism ensures that even as the primal policy becomes too safe, the dual policy continually supplies the necessary critical samples to maintain an accurate feasibility function estimate.

**Strengths:**

- It proposes several definitions to formulate an important problem in safe RL and also provides an algorithm to address it.
- It has both theoretical works and empirical experiments.

**Weaknesses:**

- There is a gap between the paper's theoretical foundation and its algorithmic implementation. Theorem 1 is built upon a Monte Carlo estimate of the feasibility function. However, the proposed FDPI algorithm is based on SAC and uses a TD estimate.

- The paper frames the "safety paradox" as a fundamental and general obstacle in safe RL. However, the entire analysis is built upon a specific problem formulation. It is unclear whether this paradox, as defined, holds for other significant branches of safe RL.

- The proposed solution, FDPI, hinges on training a dual policy that is explicitly optimized to maximize constraint violations. This approach may be impractical and dangerous for many real-world, safety-critical applications.

- The IS ratio is defined as a sequential product of policy probabilities, which is prone to high variance. To address it, the algorithm has a KL divergence constraint. However, the hyperparameter chosen for the KL divergence is high (5.0), suggesting the algorithm could be unstable.

**Questions:**

How are the hyperparameters chosen, such as the feasibility threshold (0.1) and the dual threshold (0.95)?

---

> ### Author Response · Authors · 2025-11-23
> **Authors' Response (Part 1)**
>
> We sincerely appreciate the reviewer's time and effort in reviewing our paper and providing such constructive feedback. Below, we address each of the concerns in detail.
>
> ## Weaknesses
>
> > There is a gap between the paper's theoretical foundation and its algorithmic implementation. Theorem 1 is built upon a Monte Carlo estimate of the feasibility function. However, the proposed FDPI algorithm is based on SAC and uses a TD estimate.
>
> We thank the reviewer for this insightful observation. The remark at the end of Section 4 notes that our analysis, while based on MC estimate, extends to TD estimate. The core reason is that the increased variance identified in Theorem 2 propagates to the TD target.
>
> Specifically, Section 4.2 proves that the variance of steps to violation, $N(\tau)$, increases with safer policies. Consider an initial state $x$, its subsequent state $x'$, and the sub-trajectory $\tau'$ starting from $x'$. Since $N(\tau')=N(\tau)-1$, the variance of $N(\tau')$ also increases with policy safety. This implies a higher variance in the true feasibility value $F^\pi(x')$. Consequently, the TD target, which is computed by $F^\pi(x')$, inherits this increased variance, leading to a larger estimation error in $\hat{F}^\pi(x)$.
>
> We acknowledge that this connection could have been made more explicit. We have revised the remark at the end of Section 4 and added a detailed explanation to the Appendix D.2 to clarify this point.
>
> > The paper frames the "safety paradox" as a fundamental and general obstacle in safe RL. However, the entire analysis is built upon a specific problem formulation. It is unclear whether this paradox, as defined, holds for other significant branches of safe RL.
>
> We thank the reviewer for this important question. While our analysis is based on MC estimate and the CDF, it can be extended to more general settings. In the reply to the previous point, we have explained the extension to TD estimate. Here, we demonstrate the extension to the cost value function (CVF) widely used in CMDP, defined as $F^\pi(x)=\mathbb{E}\_{\tau\sim\pi}[\sum_{t=0}^\infty \gamma^t c(x_t)|x_0=x]$.
>
> The key insight is that the CVF can be decomposed into a discounted sum of CDF-like terms. Specifically, for any trajectory, we can segment it at constraint-violating states. The CVF is then equivalent to the discounted sum of the CDFs of these segmented sub-trajectories. Since our analysis shows that the estimation error bound increases for each of these constituent CDFs as the policy becomes safer, the error bound of the total CVF estimate must also increase.
>
> This confirms that the safety paradox not only holds for our specific problem, but also applies to more general safe RL formulations. We have added this generalization to the remark at the end of Section 4 and Appendix D.2.
>
> > The proposed solution, FDPI, hinges on training a dual policy that is explicitly optimized to maximize constraint violations. This approach may be impractical and dangerous for many real-world, safety-critical applications.
>
> We thank the reviewer for raising this important concern. In the broader safe RL community, there are two training and implementation modes: (1) offline training and online deployment (OTOD), which first trains a policy in simulator and then deploys it in the real world, and (2) simultaneous online training and deployment (SOTD), which directly interacts with the real world to collect data for training. The OTOD mode only requires the final policy to be safe because intermediate policies will not be deployed in the real world. The SOTD model requires both the final policy and all intermediate policies to be safe.
>
> FDPI, along with the baselines we compare in the paper, belongs to the OTOD mode. We focus primarily on learning a safe policy at convergence, rather than guaranteeing safety during training. Therefore, FDPI does not involve unsafe exploration in the real world during training.
>
> We fully agree that for the SOTD mode, one must ensure safety throughout training. Achieving this goal would require integrating additional safe exploration techniques like those in [3,4]. These methods typically employ a model of the environment, either known or learned. Constraint violations are allowed in the model but not allowed in the environment. They alternate between learning a safe policy within the current model and refining the model with newly collected data. These approaches are complementary to our contribution, which focuses on solving the safe policy within a fixed model/environment.
>
> [3] Berkenkamp, F., et al. (2017). Safe model-based reinforcement learning with stability guarantees. Advances in neural information processing systems, 30.
>
> [4] Yu, D., et al. (2023). Safe model-based reinforcement learning with an uncertainty-aware reachability certificate. IEEE Transactions on Automation Science and Engineering, 21(3), 4129-4142.

---

> ### Author Response · Authors · 2025-11-23
> **Authors' Response (Part 2)**
>
> > The IS ratio is defined as a sequential product of policy probabilities, which is prone to high variance. To address it, the algorithm has a KL divergence constraint. However, the hyperparameter chosen for the KL divergence is high (5.0), suggesting the algorithm could be unstable.
>
> We thank the reviewer for raising this important point regarding the KL divergence threshold. The reviewer is correct that a KL value of 5.0 is considered large in standard RL algorithms like TRPO. However, the role and interpretation of this constraint in our algorithm are fundamentally different.
>
> In typical trust-region methods (e.g., TRPO, PPO), a small KL bound (e.g., 0.01-0.1) is necessary because it constrains two successive versions of the same policy, which share an identical optimization objective. In contrast, our KL constraint regulates two distinct policies with opposing objectives: the primal policy (optimizing for reward and safety) and the dual policy (explicitly seeking constraint violations). If the KL threshold were set too small, it would severely restrict the dual policy's ability to deviate from the primal policy. This would prevent the collection of sufficient violating states, thereby failing to address the safety paradox.
>
> Furthermore, from an empirical perspective, we conducted all experiments with 5 random seeds. The resulting learning curves (provided in Figure 2) do not exhibit significant oscillations or high variance, demonstrating that the chosen KL value of 5.0 is indeed practical and does not lead to instability in practice.
>
> ## Questions
>
> > How are the hyperparameters chosen, such as the feasibility threshold (0.1) and the dual threshold (0.95)?
>
> We thank the reviewer for this question. First of all, it is worth mentioning that these hyperparameters were not extensively tuned via grid search and were held constant across all environments, demonstrating their robustness.
>
> The feasibility threshold $\epsilon$ is designed to be a small value. A principled method for choosing $\epsilon$ is to set it to $\gamma^{N_0}$, where $N_0$ is the number of steps beyond which a future constraint violation is considered irrelevant. This is justified by the common assumption that the steps to violation of infeasible states are uniformly bounded [1]. Under this assumption, setting $\epsilon=\gamma^{N_0}$ still guarantees an exact recovery of the feasible region. In practice, with $\gamma=0.99$, $\epsilon=0.1$ corresponds to a horizon $N_0 \approx 229$, which is a conservative and effective choice in most environments.
>
> For the choice of the dual threshold $d$, we sincerely thank the reviewer for helping us identify a misstatement in the paper. The dual threshold should be interpreted as the threshold for the proportion of feasible states, not the average feasibility value. We have fixed the related statement in Section 5.1 and the pseudocode Appendix B. This hyperparameter directly regulates the desired balance of feasible states in the replay buffer. When the proportion of feasible states rises above $d$, the dual policy is activated to collect more infeasible states. Once the proportion falls back below $d$, the dual policy is deactivated. A value of $d=0.95$ aims to maintain a reasonable amount of infeasible states.
>
> We have also added sensitivity analyses for these two hyperparameters in Appendix C.2.
>
> [1] G. Thomas, et al. (2021). Safe reinforcement learning by imagining the near future. Advances in Neural Information Processing Systems, 34:13859–13869.

---

### Author Response · Authors · 2025-12-03

Dear Area Chair and Reviewers,

Thank you for your engagement in reviewing our paper and for providing the valuable feedback. We are encouraged that four of our six reviewers initially recommended acceptance, with scores of 8, 6, 6, and 6. After the rebuttal, one of the 4 score reviewers (opKu) would raise the score, and one of the 6 score reviewers (HJpx) have raised the score.

A major concern from the two 4 score reviewers (EGeQ and opKu) is the scope of our theory. Specifically, they comment that our theory is built upon the CDF and Monte Carlo estimate. However, we demonstrate in our rebuttal that our theory can be naturally extended to broader settings including:

- The **cost value function** widely used in CMDP by decomposing it to CDF-like terms, and
- **TD estimate** by observing that the increased variance propagates to the TD target.

The explanation for these extensions has been added to the paper in Appendix D.2.

Another concern from the two 4 score reviewers is the training-time violation of our algorithm. We believe that this is based on a misunderstanding of our problem setting. In our rebuttal, we clarify that we focus on the setting of training in simulator before deployment in real world, and thus training-time violation is allowed.

In addition, we also improved the paper following other comments from the reviewers:

- HJpx, 2igd, 3DTX: Add sensitivity analyses for hyperparameters.
- HJpx, 2igd, 3DTX: Improve the justification for assumptions.
- HJpx, W7n2: Add discussion and comparison to related works.
- opKu: Add experiments on a new task.

We thank you again for your time and consideration.

---

### Meta-Review · Area_Chair_WMkq · 2025-12-22

**Summary:**

**Paper Summary**: This paper considers the safe RL problem where one needs to ensure that the feasibility constraints are satisfied. The paper shows one important contribution where improving policy safety reduces the frequency of constraint-violating samples, thereby impairing efficient function approximation, resulting in a potential performance degradation. In order to address this problem, the paper proposed to introduce a dual policy that will try to maximize the constraint violation if the ratio of the violating samples and non-violating samples becomes very less in order to accelerate the learning procedure. The empirical results show the improvement over the baselines, including the feasible policy iteration (FPI) algorithm.

**Reviewers' Concerns**: Reviewers appreciate the overall contribution of the paper. But, they also raised a few concerns. For example, the reviewer EGeQ pointed out that the paper itself considers an importance sampling procedure, which might have a higher variance, resulting in a large number of samples, negating their proposed goals. The reviewer also raised the theoretical contribution, as it is based on the Monte-Carlo setup. Multiple reviewers have also raised concerns about the lack of baselines and hyperparameter sensitivity analysis.

**AC's take**: The AC believes that the authors' rebuttals have addressed those concerns. The AC would urge the authors to include the discussions and the new results in the main paper. The AC would also urge the authors to consolidate the claim that the proposed approach is applicable to the broader class, including the CMDP setup.

**Reviewer Concerns:**

Reviewers mostly appreciate the contributions. While most of the reviewers have stated that their concerns have been addressed. Reviewer EGeQ did not respond, but, based on the responses, it seems that the reviewer's concerns are addressed.

**Reviewer Scores:**

This paper has 6 reviewers, out of which two have scored negatively (4). One of the reviewers responded that they are satisfied and are going to increase the score. The other reviewer did not respond, but based on the rebuttals, it is clear that the reviewer's concerns are addressed.

---

### Decision · Program_Chairs · 2026-01-26

Accept (Poster)